# Kupffer cell and recruited macrophage heterogeneity orchestrate granuloma maturation and hepatic immunity in visceral leishmaniasis

Gabriela Pessenda[1], Tiago R. Ferreira [1], Andrea Paun[1], Juraj Kabat[2], Eduardo P. Amaral [1,3], Olena Kamenyeva [2], Pedro Henrique Gazzinelli-Guimaraes [1,4], Shehan R. Perera[5], Sundar Ganesan[2], Sang Hun Lee[1] & David L. Sacks [1] ✉

In murine models of visceral leishmaniasis (VL), the parasitization of resident Kupffer cells (resKCs) drives early *Leishmania infantum* growth in the liver, leading to granuloma formation and subsequent parasite control. Using the chronic VL model, we demonstrate that polyclonal resKCs redistributed to form granulomas outside the sinusoids, creating an open sinusoidal niche that was gradually repopulated by monocyte-derived KCs (moKCs) acquiring a tissue specific, homeostatic profile. Early-stage granulomas predominantly consisted of CLEC4F+KCs. In contrast, late-stage granulomas led to remodeling of the sinusoidal network and contained monocyte-derived macrophages (momacs) along with KCs that downregulated CLEC4F, with both populations expressing iNOS and pro-inflammatory chemokines. During late-stage infection, parasites were largely confined to CLEC4F-KCs. Reduced monocyte recruitment and increased resKCs proliferation in infected *Ccr2*$^{-/-}$ mice impaired parasite control. These findings show that the ontogenic heterogeneity of granuloma macrophages is closely linked to granuloma maturation and the development of hepatic immunity in VL.

Kupffer cells (resKCs) are embryonically derived, tissue-resident macrophages (TRMs) that reside within the liver sinusoids and remain non-migratory under homeostatic conditions[1–5]. In certain inflammatory contexts or when selective depleted, they can be replaced by monocyte-derived KCs (moKCs)[6–17]. In mice, KCs are conventionally identified as F4/80$^{hi}$CD11b$^{int}$ cells[18], and the acquisition of KC-specific transcription factors and markers by monocyte-derived macrophages (momacs) is typically used to define their differentiation into KCs.

The C-type lectin CLEC4F and the mucin-domain containing protein TIM-4 have been identified as KC markers[6–8,12,19,20]. Selective depletion of KCs using diphtheria toxin (DT) leads to extensive monocyte infiltration into the liver and their subsequent differentiation into KCs. During this process, KC-specific transcription factors

[1]Laboratory of Parasitic Diseases, National Institute of Allergy and Infectious Diseases, National Institutes of Health, Bethesda, MD, USA. [2]Biological Imaging Section, Research Technology Branch, National Institute of Allergy and Infectious Diseases, National Institutes of Health, Bethesda, MD, USA. [3]Inflammation and Innate Immunity Unit, Laboratory of Clinical Immunology and Microbiology, National Institute of Allergy and Infectious Diseases, National Institutes of Health, Bethesda, MD, USA. [4]Department of Microbiology, Immunology & Tropical Medicine School of Medicine & Health Sciences. The George Washington University, Washington DC, USA. [5]Department of Electrical and Computer Engineering, The Ohio State University, Columbus, OH, USA. ✉e-mail: dsacks@niaid.nih.gov

and F4/80 are rapidly upregulated, while CLEC4F, and particularly TIM-4, show delayed expression[6,7,12]. Consequently, these markers can be used, at least in the shorter term, to distinguish the ontogenic origin of these cells.

Granulomas are organized aggregates of macrophages and other immune cells that form in response to infectious or non-infectious agents[21]. Experimental visceral leishmaniasis (VL) caused by *Leishmania infantum* and *L. donovani* is a well-characterized chronic infection model for studying granuloma dynamics and function in the liver[22]. Granuloma formation is initiated around parasitized KCs, which cluster and become encircled by innate immune cells and T lymphocytes[22–27]. The granuloma structure, along with pro-inflammatory cytokines secreted by its constituent cells, creates a microenvironment that contains and ultimately clears the infection from the liver[27]. The possible heterogeneity of granuloma macrophages in terms of their ontogeny and function remains unexplored.

Here, we provide direct evidence for a striking liver macrophage heterogeneity during experimental VL, informing macrophage ontogeny, spatial distribution, and function in granuloma maturation and immunity. Late-stage VL results in two distinct hepatic compartments, each populated by unique macrophage subsets. The sinusoidal compartment comprises resKCs and regulatory moKCs. The mature granuloma compartment, located outside of the sinusoids, contains pro-inflammatory activated macrophages, including both infected iNOS+CLEC4F−KCs and uninfected iNOS+ momacs. Parasite control was significantly compromised in the absence of monocyte recruitment. Our findings show that VL induces remodeling of the perisinusoidal space by granulomas containing ontogenically heterogeneous macrophage subsets, which play a key role in parasite control in the liver.

## Results

### Heterogeneity of F4/80+ macrophages during VL defined by the differential expression of CLEC4F and TIM-4

Kupffer cells in naïve mice are characterized as F4/80hiCD11bintCD64+CLEC4F+TIM-4+ (Supplementary Fig. 1a). At 42 days post *L. infantum* infection (42 d.p.i.), we identified 4 subsets of F4/80hiCD11bintCD64+ liver macrophages in C57BL/6 mice. The frequency of CLEC4F+TIM-4−resKCs was reduced, while the proportions of CLEC4F+TIM-4−moKCs, and CLEC4F−TIM-4+ and CLEC4F−TIM-4− macrophages were increased compared to naïve mice (Supplementary Fig. 1b–d). To investigate these subsets in situ, we performed confocal microscopy on liver sections at multiple time points (Fig. 1a). In naïve and 19 d.p.i. mice ("early-stage infection"), the majority of F4/80+ macrophages were CLEC4F+TIM-4+resKCs. By 42 d.p.i. ("late-stage infection"), these cells were significantly reduced in frequency and number, coinciding with a peak in the CLEC4F−TIM-4+ and CLEC4F−TIM-4− populations. CLEC4F+TIM-4−moKCs continued to accumulate during the "resolving phase" at 72 d.p.i. (Fig. 1a, b). Consistent with whole-liver observations, analysis of the cellularity within the granulomas revealed that CLEC4F+TIM-4+resKCs were the principal subset found within early-stage granulomas at 19 d.p.i. (Fig. 1c, d). However, by 42 d.p.i., these cells were significantly reduced in granulomas, which were predominantly populated by CLEC4F−TIM-4+ and CLEC4F−TIM-4− cells. By 72 d.p.i., granulomas, while few in number, contained relatively equal proportions of all four F4/80+ subsets (Fig. 1c, d). Overall, resKCs were markedly reduced during late VL, in favor of F4/80+ macrophages lacking CLEC4F and/or TIM-4 expression. While early-stage granulomas were largely composed of resKCs, later infection resulted in granulomas containing heterogeneous macrophage populations.

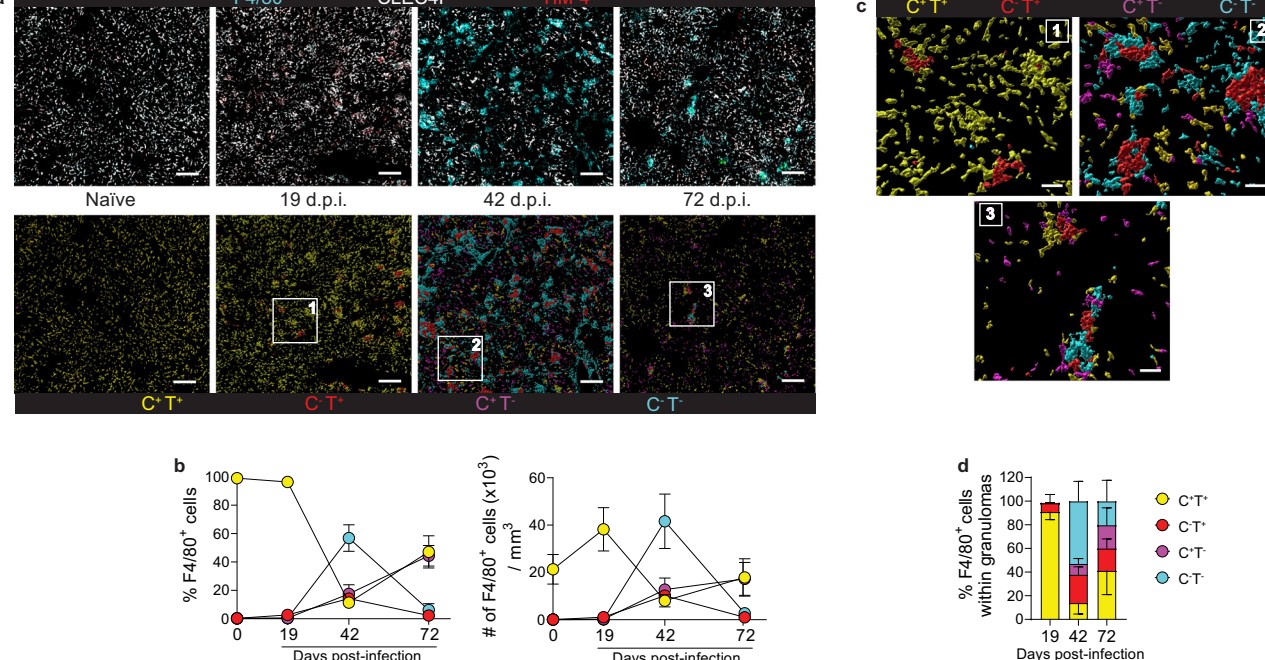

**Fig. 1 | CLEC4F and TIM-4 serve as markers for identifying macrophage heterogeneity in VL. a** Top: Representative confocal microscopy images showing wild-type naïve, 19-, 42-, and 72-day infected livers stained with anti-F4/80 (cyan), anti-CLEC4F (white), and anti-TIM-4 (red). Bottom: Surface rendered F4/80+ cells classified according to their CLEC4F and TIM-4 expression, CLEC4F+TIM-4+ (yellow), CLEC4F−TIM-4+ (red), CLEC4F+TIM-4− (magenta), and CLEC4F−TIM-4− (cyan). Scale bars, 200 μm. **b** Frequency and number of F4/80+ cells classified according to their CLEC4F and TIM-4 expression, in naïve and infected mice at different days post-infection. Data pooled from 2 independent experiments (n = 6 for naïve and n = 7 for infected mice). **c** Granulomas identified in (**a**) are shown in the inset and were defined as clusters of F4/80+ cells with volumes greater than 1.03 × 10⁴ μm³. Scale bars, 40 μm. **d** Proportion of CLEC4F+ and/or TIM-4+ cells in F4/80+ granulomas at different times post-infection. Data pooled from 2 (19- and 72 d.p.i.) and 4 (42 d.p.i.) independent experiments (n = 7 for 19- and 72 d.p.i. and n = 15 for 42 d.p.i.). Values in (**b** and **d**) represent the mean ± SD. Source data are provided as a Source Data file.

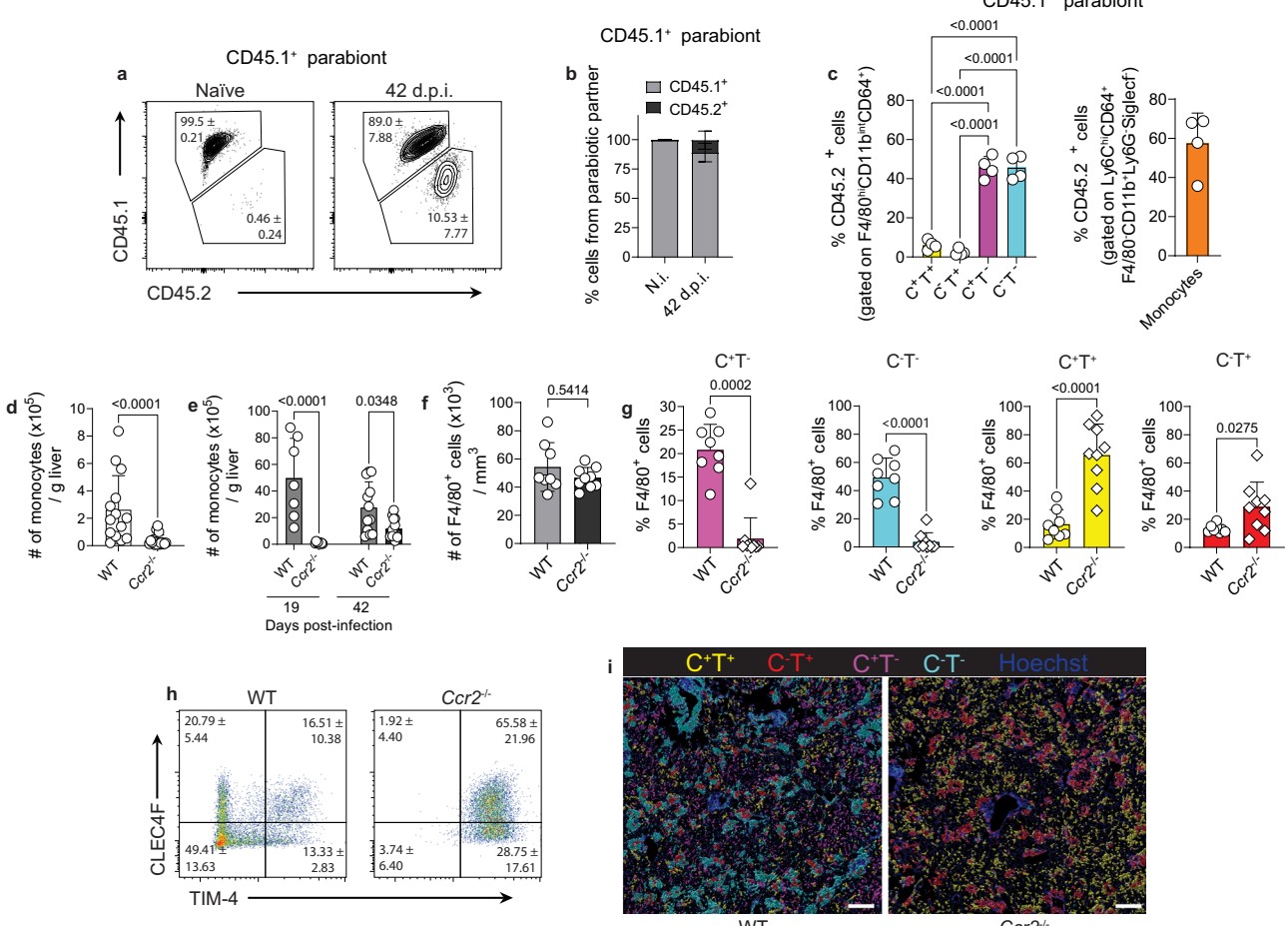

**Fig. 2 | TIM-4⁻ macrophages are monocyte-derived during late-stage VL.**
**a** Representative contour plots showing the percentages of chimerism in F4/80⁺CD11bⁱⁿᵗCD64⁺ cells from naïve and 42-day infected CD45.1⁺ parabiont.
**b** Percentages of chimerism in F4/80⁺CD11bⁱⁿᵗCD64⁺ cells from uninfected and infected CD45.1⁺ parabiotic partners at 42 d.p.i. **c** Frequency of CD45.2⁺ cells in infected CD45.1⁺ parabionts, gated on F4/80⁺CD11bⁱⁿᵗCD64⁺CLEC4F⁺/⁻TIM-4⁺/⁻ macrophages or Ly6C⁺CD11b⁺Ly6G⁻SiglecF⁻ monocytes. Data pooled from two independent experiments (*n* = 3 for uninfected and *n* = 4 for infected pairs).
**d**, **e** Number of monocytes in the livers of naïve (**d**) and 19- and 42-day infected WT and *Ccr2⁻/⁻* mice (**e**) determined by flow cytometry. Data pooled from 5 independent experiments (naïve, n = 14 for WT and n = 17 for *Ccr2⁻/⁻*; 19 d.p.i., n = 7 for WT and n = 9 for *Ccr2⁻/⁻*; 42 d.p.i., n = 12 for WT and n = 14 for *Ccr2⁻/⁻*). **f** Number of F4/80⁺ cells in WT and *Ccr2⁻/⁻* at 42 d.p.i., quantified from confocal microscopy images. **g** Frequency of macrophage subsets based on CLEC4F and TIM-4 expression in WT

and *Ccr2⁻/⁻* mice at 42 d.p.i., quantified from confocal microscopy images. Data pooled from 2 independent experiments (*n* = 8 for WT and *n* = 9 for *Ccr2⁻/⁻*). In (**b**–**g**) values show mean ± SD. In (**d**, **f**) for data that passed the normality test, *P* values were obtained using a two-tailed unpaired *t* test. For data that did not pass the normality test, *P* values were obtained using a two-tailed Mann-Whitney test. In (**c**) *P*-values were obtained using ordinary one-way ANOVA with Tukey's multiple comparisons, in (**e**) *P*-values were obtained using ordinary one-way ANOVA with Sidak's multiple comparisons. **h** Representative dot plots from confocal microscopy images showing the frequency of macrophage subsets in infected WT and *Ccr2⁻/⁻* mice at 42 d.p.i. Numbers indicate mean ± SD percentage of cells in the gate. **i** Representative rendered confocal microscopy images of 42 d.p.i. livers from WT and *Ccr2⁻/⁻* mice showing CLEC4F⁺TIM-4⁺ (yellow) and CLEC4F⁻TIM-4⁺ (red) macrophages, CLEC4F⁺TIM-4⁻ (magenta) moKCs and CLEC4F⁻TIM-4⁻ (cyan) momacs. Scale bars, 200 μm. Source data are provided as a Source Data file.

## TIM-4⁻ macrophages are monocyte-derived and KCs downregulate CLEC4F in late-stage granulomas

To confirm the origin of the macrophage subsets identified in the liver at 42 d.p.i., we generated congenically paired parabiotic mice, which share a chimeric blood supply. In naïve mice, nearly all F4/80ʰⁱCD11bⁱⁿᵗCD64⁺ macrophages carried the congenic marker of the host parabiont, corroborating their identity as TRMs. At 42 d.p.i., 10.5 ± 7.8% of the cells originated from the congenic partner (Fig. 2a, b). Further classification of CD45.2⁺ cells within infected livers of the CD45.1 parabiont revealed that TIM-4⁻ macrophages and monocytes contained cells from the congenic partner (Fig. 2c). In contrast, TIM-4⁺ macrophages were primarily from the host parabiont (Fig. 2c). These findings provide strong evidence that the absence of TIM-4 expression identifies macrophages of monocytic origin during VL.

We also used *Ccr2⁻/⁻* mice, in which recruitment of *Ccr2⁺* monocytes to the liver was markedly reduced in both naïve (Fig. 2d) and

infected mice (Fig. 2e). Confocal microscopy showed that although F4/80⁺ macrophage numbers were comparable between infected WT and *Ccr2⁻/⁻* mice (Fig. 2f), the frequency of TIM-4⁻ subsets were significantly reduced in *Ccr2⁻/⁻* mice (Fig. 2g). Conversely, the proportions of TIM-4⁺ subsets were increased compared to WT mice (Fig. 2g). These changes reflected overall differences in macrophage heterogeneity, with infected WT mice exhibiting all four F4/80⁺ subsets, whereas infected *Ccr2⁻/⁻* mice predominantly showed TIM-4⁺ macrophages (Fig. 2h, i and Supplementary Fig. 2a).

It has been reported that residual monocyte infiltration in *Ccr2⁻/⁻* livers is enough to repopulate an open KC niche due to increased proliferation of the few engrafted monocyte-derived cells[6]. In *L. infantum*-infected WT mice, around 60% of the proliferating cells were F4/80⁻ (Supplementary Fig. 2b). Among the proliferating F4/80⁺ cells, the majority were the CLEC4F⁻TIM-4⁻ momacs, while resKCs corresponded to 8%, and CLEC4F⁻TIM-4⁺ cells to 30% of the proliferating

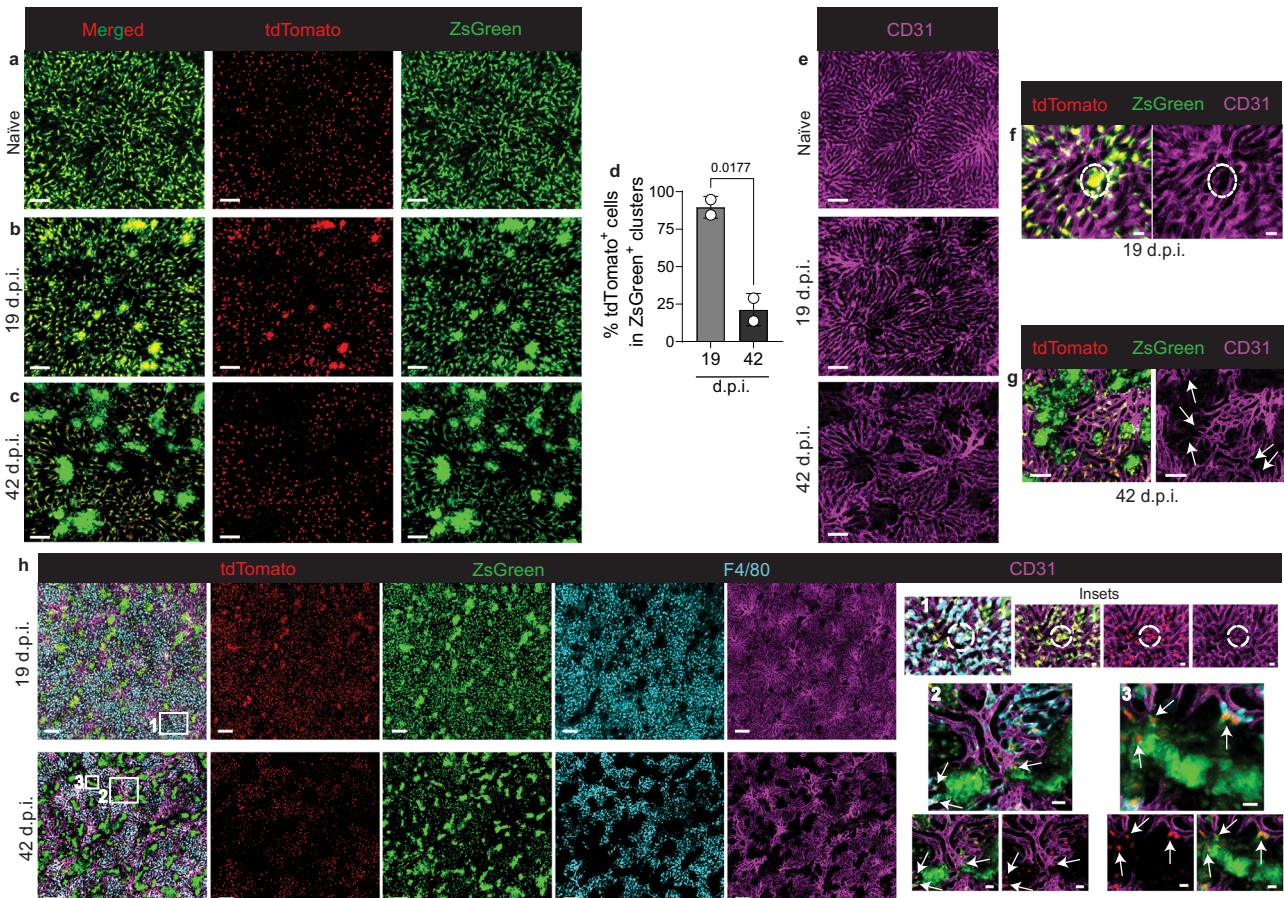

**Fig. 3 | Kupffer cells in late-stage granulomas are CLEC4F⁻ and are located outside the sinusoids. a–c** Representative live images from naïve (**a**), 19-day (**b**) and 42-day (**c**) infected *Clec4f^Cre-TdT^ZsGreen* mice, showing *Clec4f* active expression (red) and *Clec4f* previous expression (green). Scale bars, 100 μm. **d** Frequency of tdTomato⁺ cells identified in ZsGreen⁺ granulomas at 19 and 42 d.p.i. Data pooled from live imaging carried out in 4 independent experiments (*n* = 2 for 19 d.p.i. and *n* = 2 for 42 d.p.i.). Values represent mean ± SD. *P*-values were obtained using a two-tailed unpaired *t* test. **e** Sinusoid distribution from images (**a–c**), visualized by CD31 staining (magenta). Scale bars, 100 μm. **f** Representative images showing a tdTomato⁺ZsGreen⁺ cluster (red and green) and the sinusoids (magenta) at 19 d.p.i.

Scale bars, 20 μm. **g** Representative images showing tdTomato⁻ZsGreen⁺ clusters (green) and the sinusoids (magenta) at 42 d.p.i. Scale bars, 50 μm. **h** Images of *Clec4f^Cre-TdT^ZsGreen* mice at 19 and 42 d.p.i., showing *Clec4f* active expression (red), *Clec4f* previous expression (green), F4/80 (cyan), and sinusoids (magenta). Scale bars, 200 μm. Inset 1, the dotted line indicates a small cluster of F4/80⁺tdTomato⁺ZsGreen⁺KCs in contact with intact sinusoids. Scale bars, 20 μm. Insets 2 and 3 show tdTomato⁻ZsGreen⁺ clusters outside the sinusoids. Arrows point to tdTomato⁺ZsGreen⁺ F4/80⁺ or F4/80⁻ KCs within granulomas, next to or in contact with the sinusoids. Scale bars, 15 μm. Source data are provided as a Source Data file.

cells (Supplementary Fig. 2c, d). Thus, proliferation might support increased frequencies of CLEC4F⁻TIM-4⁺ cells within WT granulomas. In infected *Ccr2^−/−^* mice, 60% of the proliferating cells were again F4/80⁻ (Supplementary Fig. 2b), but proliferation was increased 4.6-fold in resKCs, and 1.7-fold in CLEC4F⁻TIM-4⁺ cells compared to WT mice (Supplementary Fig. 2c, d), presumably to compensate the reduced monocyte infiltration in these mice.

The parabiosis experiments showing that the CLEC4F⁻TIM-4⁺ cells were host-derived, in addition to their presence in granulomas of the *Ccr2^−/−^* mice, suggested that these cells could be KCs that downregulated CLEC4F expression within granulomas. Breeding *Clec4f^Cre-TdT^* mice with *RCL-ZsGreen* mice, which have a *loxP*-flanked STOP cassette preventing transcription of CAG promoter-driven *ZsGreen*, allowed us to distinguish CLEC4F active expression, marked by tdTomato, from its prior expression, indicated by ZsGreen. Live imaging of naïve *Clec4f^Cre-TdT^ZsGreen* mice confirmed that KCs were tdTomato⁺ZsGreen⁺ (Fig. 3a). At 19 d.p.i., some clusters of tdTomato⁺ZsGreen⁺KCs were observed (Fig. 3b, d). By 42 d.p.i, tdTomato⁺ZsGreen⁺KCs appeared as individual cells, while KC clusters were predominantly tdTomato⁻ZsGreen⁺ (Fig. 3c, d). These results confirm that CLEC4F⁻TIM-4⁺ cells are KCs that downregulate CLEC4F expression in late-stage granulomas.

Taken together, infected livers exhibited an ontogenically heterogeneous macrophage population consisting of CLEC4F⁺TIM-4⁺resKCs and CLEC4F⁻TIM-4⁺KCs, while CLEC4F⁺TIM-4⁻moKCs and CLEC4F⁻TIM-4⁻momacs were monocyte-derived and absent in infected *Ccr2^−/−^* livers.

## Late-stage granulomas remodel the sinusoidal network without causing vascular damage

Previous VL studies have shown that KCs can redistribute to form granulomas[23], and based on H&E staining it was suggested that granuloma expansion causes loss of the physical association between KCs and the sinusoids[26]. These studies did not directly observe the sinusoidal network and could not determine whether KCs redistribution to form granulomas occurred inside or outside the sinusoids. By using CD31 to label the sinusoids in *Clec4f^Cre-TdT^ZsGreen* mice, we observed that *L. infantum* infection caused remodeling of the sinusoidal network (Fig. 3e). At 19 d.p.i., small early clusters were identified partially outside the sinusoids, but they did not alter sinusoid distribution or integrity (Fig. 3f and Supplementary Movie 1). At 42 d.p.i., changes in the sinusoidal network became evident around large ZsGreen clusters, which were mostly outside the blood vessels (Fig. 3g and Supplementary movie 2). The sinusoids appeared to be displaced, pushed

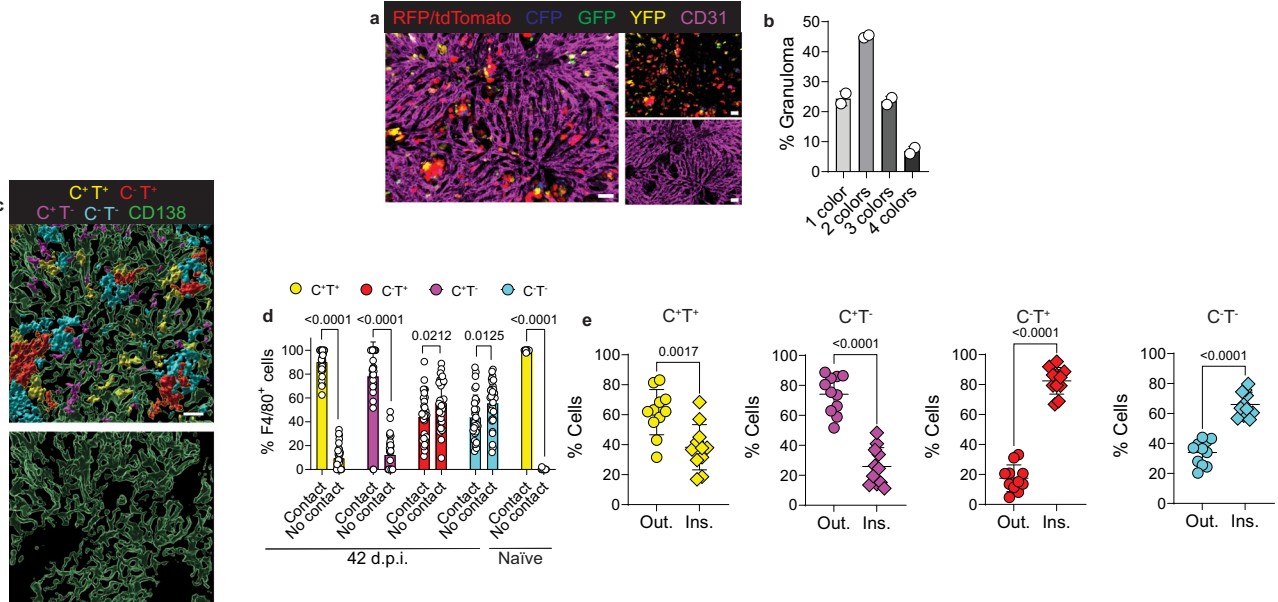

**Fig. 4 | Clonality and spatial distribution of macrophage subsets during late-stage VL. a** Intravital microscopy images representing a 42-day infected liver from a *Clec4f^{Cre-TdT}Confetti* mouse, stained with anti-CD31 (magenta), and showing tdTomato (red-nucleus), RFP (red-cytoplasm), YFP (yellow-cytoplasm), GFP (green-nucleus), and CFP (blue-membrane) KCs. Scale bars, 50 μm. **b** Frequency of KC clones in granulomas based on KC colors. Data pooled from live imaging performed in 2 independent experiments (*n* = 2 and quantification of 284 granulomas). **c** Representative rendered immunofluorescence image of a 42-day infected, wild-type liver showing the spatial distribution of CLEC4F^+TIM-4^+resKCs (yellow), CLEC4F^-TIM-4^+KCs (red), CLEC4F^+TIM-4^- moKCs (magenta), CLEC4F^-TIM-4^-momacs (cyan), and sinusoids (green). Scale bars, 30 μm. **d** Bar graphs showing the localization of each macrophage subset according to

their interaction with the sinusoids in 42-day infected and naïve, wild-type mice. Data pooled from 2 independent experiments using 4 naïve mice and 8 infected mice (42 d.p.i.). Frequencies were calculated from four regions of interest (ROIs) per infected mouse (*n* = 32 ROIs) and 2-3 ROIs per uninfected mouse (*n* = 11 ROIs). **e** Scatter plots from immunofluorescence images showing the frequency of each F4/80^+ population based on their distribution outside or inside granulomas at 42 d.p.i. Data pooled from 3 independent experiments (*n* = 11). In (**d, e**) for data that passed the normality test, *P*-values were obtained using a two-tailed unpaired *t* test. For data that did not pass the normality test, *P*-values were obtained using a two-tailed Mann-Whitney test. Values from (**b, d, e**) represent mean ± SD. Source data are provided as a Source Data file.

outward and downward by the expanding granuloma in the perisinusoidal space (Fig. 3g, arrows, and Supplementary Movie 3). Importantly, there were no signs of damage to the vessel walls or changes in blood vessels diameter (Fig. 3g, arrows, and Supplementary Movie 3).

To further investigate if the formation of KC clusters was associated with damage or rupture to the sinusoids, we stained red blood cells (RBCs) with Ter119 and performed live imaging in WT mice at 3- and 6-w.p.i. RBCs were observed flowing within the sinusoids, however, no blood flow was detected in areas where granulomas were present (Supplementary Movies 4–7). Consistently, tracking of RBCs revealed their movement within the sinusoids but not in regions containing granulomas (Supplementary Fig. 3a). We did, however, detect RBCs within some F4/80^+ clusters at 3 w.p.i. (Supplementary Movies 4 and 5) and in areas corresponding to granulomas at 6 w.p.i. (Supplementary Movies 6 and 7). However, unlike the dynamic streaming of RBCs observed within the sinusoids, these RBCs were mostly stationary. In addition, static RBCs were observed in association with several F4/80^+ cells inside the sinusoids (Supplementary Movie 7).

Intravascular staining allows the distinction between tissue and blood-borne cells[28]. By intravenous (i.v.) administration of anti-F4/80, we detected small F4/80^+ tdTomato^+ZsGreen^+ clusters at 19 d.p.i., suggesting these cells remained at least partially in contact with the sinusoids/circulating blood (Fig. 3h, inset 1). By contrast, larger ZsGreen^+ clusters at 19- and 42-d.p.i. were not stained following i.v. anti-F4/80 administration (Fig. 3h), suggesting a loss of contact with the sinusoids and the absence of blood flow into the perisinusoidal space. When detected inside granulomas, F4/80^+ and/or tdTomato^+ZsGreen^+KCs were located closer to the sinusoids (Fig. 3h, arrows in insets 2 and 3).

To address the clonality of the KCs within granulomas, we bred *Clec4f^{Cre-TdT}* mice with *R26R-Confetti* mice in which Cre recombinase causes permanent expression of one of four possible fluorescent proteins, allowing us to trace clonal lineages. At 42 d.p.i., ~ 70% of the granulomas contained two or more colors of KCs (Fig. 4a, b). Thus, most granulomas were formed by the redistribution of polyclonal KCs and were not solely aggregates of self-proliferating KCs.

Live imaging of *Clec4f^{Cre-TdT}ZsGreen* and *Clec4f^{Cre-TdT}Confetti* infected livers allowed us to track KCs, but it did not discriminate resKCs from moKCs, both of which are CLEC4F^+, or CLEC4F^-TIM-4^-momacs within granulomas. Using confocal microscopy on 42 d.p.i. wild-type livers, we showed that CLEC4F^+TIM-4^+resKCs and CLEC4F^+TIM-4^-moKCs maintained their contact with the sinusoids. In contrast, nearly 60% of the CLEC4F^-TIM-4^+KCs and CLEC4F^-TIM-4^-momacs had lost contact and were located outside the sinusoids (Fig. 4c, d). Consistently, most CLEC4F^-KCs and momacs were observed within granulomas, while resKCs and moKCs were predominantly outside granulomas (Fig. 4e).

Taken together, our findings suggest that resKCs cross the sinusoidal endothelium to form granulomas in the perisinusoidal space, creating an open niche that is subsequently filled by moKCs. As granulomas expand and other cells, such as momacs, surround the KC core, the sinusoidal network is displaced, leading to resKCs losing their contact with the sinusoids and downregulating their CLEC4F expression.

## BACH1 regulates resKC proliferation and macrophage lipid peroxidation during VL

The replacement of resKCs by moKCs may also occur due to KC niche availability following cell death[8,9,15,17,29]. Murine models of *Listeria*

*monocytogenes*[9] infection and viral hepatitis[11] have demonstrated a massive loss of resKCs, with rapid replacement by moKCs. In our chronic VL model, resKCs replacement was gradual, with CLEC4F[+]moKCs occupying the sinusoidal space and coexisting with remaining resKCs at 42 d.p.i. To assess the potential role of cell death in our VL model, we used mice deficient in MLKL, a key protein in the necroptotic cell death pathway, and mice deficient in Caspase 1, involved in the lytic, pyroptotic cell death pathway[30]. At 42 d.p.i., no differences were observed in the frequencies of resKCs or moKCs in *Mlkl*[−/−] (Supplementary Fig. 4a) or *Casp1*[−/−] livers (Supplementary Fig. 4b). Using cleaved caspase 3 staining to detect apoptotic cells in situ at 42 d.p.i., we observed an increased number of clv-casp3[+] cells in all F4/80[+] subsets and in F4/80[−] cells, except in resKCs (Supplementary Fig. 4c). During infection, although the majority of clv-casp3[+] cells were F4/80[−], the frequency of apoptotic resKCs decreased, while clv-casp3[+] cells increased among all other F4/80[+] macrophages (Supplementary Fig. 4d, e).

Ferroptosis, an iron-dependent oxidative cell death characterized by lipid peroxidation and damage to biological membranes[31], has been implicated in various hepatic diseases, including malaria[29,32]. By flow cytometry, we found evidence of ferroptotic cell death in both TIM-4[+] and TIM-4[−] macrophages at 42 d.p.i. (Supplementary Fig. 5a). Glutathione peroxidase 4 (GPX4) mediates the reduction of phospholipid hydroperoxides using glutathione (GSH) as a co-factor[31], and lower levels of GPX4 and/or GSH are commonly associated with ferroptosis[33–36]. At 42 d.p.i., GSH levels were reduced compared to uninfected controls (Supplementary Fig. 5b). BACH1, a pro-oxidant factor that represses NRF2, a master regulator of host antioxidant responses[37,38], has been shown to reduce oxidative stress-mediated ferroptosis when deficient[38,39]. TIM-4[+] and TIM-4[−] cells from *Bach1*[−/−] mice at 42 d.p.i. exhibited reduced levels of lipid peroxidation (Supplementary Fig. 5c). Furthermore, in infected *Bach1*[−/−] mice, the frequency and number of resKCs (Supplementary Fig. 5d–f, h) and the total number of F4/80[+] macrophages (Supplementary Fig. 5g, h) were increased. In contrast, the frequencies of moKCs and momacs were reduced (Supplementary Fig. 5d, e), although their absolute numbers were the same compared to WT mice (Supplementary Fig. 5f, h). The frequency of infiltrating monocytes remained unchanged (Supplementary Fig. 5i). In infected *Bach1*[−/−], resKCs showed a 2.9-fold increase in proliferation compared to WT mice, while no difference in proliferation was observed among other F4/80[+] (Supplementary Fig. 5j) or F4/80[−] cells (Supplementary Fig. 5k). Despite the accumulation of resKCs, parasite loads did not change between WT and *Bach1*[−/−] mice (Supplementary Fig. 5l).

Collectively, apoptosis was increased during VL in F4/80[−] cells and in all F4/80[+] cells except resKCs, while evidence of ferroptosis was observed in resKCs and other F4/80[+] macrophages. In addition, the absence of BACH1 resulted in increased resKCs proliferation. The findings suggest that BACH1 negatively regulates resKC numbers through mechanisms involving reduced proliferation, increased lipid peroxidation, and potentially ferroptosis during the peak immune response at 42 d.p.i., without affecting parasite loads.

### Macrophage heterogeneity during VL revealed by single-cell RNA sequencing

To further investigate the cell heterogeneity identified by confocal microscopy and flow cytometry, we sorted F4/80[hi]CD11b[int]CD64[+] macrophages and performed single-cell RNA sequencing (scRNA-seq) on 7673 cells, of which 7352 were macrophages. The macrophage identification and nomenclature dataset from Remmerie et al.[14] was used as a reference for mapping our single-cell data onto the UMAP structure. UMAP projection and clustering revealed 3 main clusters in naïve and 6 clusters in infected mice (Supplementary Fig. 6a). Re-clustering and manual annotation identified two distinct clusters of Transitioning monocytes in infected mice (Fig. 5a). The top 10

differentially expressed genes (DEGs) for all clusters are shown in a heatmap (Supplementary Fig. 6b), and a full list of DEGs for each cluster identified in our scRNA-seq dataset is presented in Supplementary Data 1. The conserved KC transcriptomic signature described by Guilliams et al.[19] confirmed that cells in ResKCs, MoKCs, and Transitioning monocytes1 clusters expressed KC signature genes (Fig. 5b). In naïve mice, the ResKCs cluster contained only CLEC4F[+]TIM-4[+]resKCs, which lacked monocytic markers. In infected mice, the ResKCs cluster also included monocyte-derived macrophages, identified by *Ccr2/Cxc3cr1* and/or lack of *Clec4f/Timd4* expression, suggesting these were moKCs transcriptionally similar to resKCs that still had not acquired CLEC4F/TIM-4 expression (Fig. 5c, d and Supplementary Fig. 6b). Proliferating macrophages were also increased in infected compared to naïve mice (Fig. 5a, e and Supplementary Fig. 5a). Remmerie et al.[14] identified the Mac1 cluster as pre-moKCs due to *Clec1b* expression, a marker for moKCs expressed earlier than *Clec4f*[8]. We did not detect *Clec1b* or KC signature genes in cells from this cluster (Fig. 5b, d). However, given that these cells expressed several KC-associated transcription factors (Supplementary Fig. 6c), they might eventually differentiate into KCs in infected mice. The Mac2 cluster, previously identified as hepatic lipid-associated macrophages (hep-LAMS)[14], showed higher expression of *Spp1*, *Cd9*, and *Trem2*, consistent with this macrophage subtype (Fig. 5f). Finally, the term Transitioning monocytes was used to describe cells exhibiting features of both monocytes and macrophages[14]. The expression of KC signature genes and multiple KC transcription factors in the Transitioning monocytes1 cluster (Fig. 5b and Supplementary Fig. 6c) may also suggest that these macrophages were committed to a KC fate.

Of note, we could not identify a defined cluster containing CLEC4F[−]TIM-4[+] cells, which were well represented in 42-day infected livers by confocal microscopy. This absence may be due to cell recovery bias during ex vivo analysis, resulting from liver disaggregation, digestion[40], or high death rates among resident macrophages, causing an underrepresentation of resKCs and CLEC4F[−]KCs vs monocyte-derived populations. These cells could also be dispersed within the clusters identified in our scRNA-seq. To reveal their transcriptional program(s), we sorted F4/80[hi]CD11b[int]CD64[+]CLEC4F[−]TIM-4[+] cells from 42-day infected mice liver samples and performed a new scRNA-seq analysis. The list of DEGs for each cluster identified is presented in Supplementary Data 2. A pseudo bulk comparison with our previously identified clusters showed that more than half of the sorted cells had gene expression profiles similar to resKCs and proliferating cells, while around 20% were grouped in the Transitioning Monocytes clusters (Supplementary Fig. 6d). The CLEC4F[−]TIM-4[+] sorted cells shared expression of several KC-associated transcription factors, although *Nr1h3*, *Irf7*, and *Mafb* expression was lower than compared to cells in the ResKCs cluster (Supplementary Fig. 6e). In addition, cells in the ResKCs cluster and CLEC4F[−]TIM-4[+] sorted cells displayed significant transcriptional similarity regarding KC identity genes[19] (Supplementary Fig. 6f).

### Functional heterogeneity of macrophage subsets and their contribution to *L. infantum* control

Functional analysis of our scRNA-seq revealed that in infected mice, the ResKCs cluster expressed genes associated with KC functions, such as lipid and iron metabolism[19] (Fig. 6a). In addition, cells in this cluster expressed *Cxcl13* and *Ccl24* (Fig. 6b). While most resKCs retained their putative homeostatic roles, they were also responsive to the Th1 inflammatory environment of infected livers[41,42], as evidenced by their expression of *Cxcl10*, *Cxcl9* and *Il1a* (Fig. 6b, g). CLEC4F[−]TIM-4[+] cells expressed the highest levels of *Il10* and *Il1a*, and of some inflammatory chemokines, such as *Cxcl2*, *Ccl3*, and *Ccl4* (Supplementary Fig. 7a). MoKCs and Mac1 clusters showed a chemokine-cytokine profile that resembled more resKCs than macrophages in other clusters but did not express lipid and iron metabolism-associated genes (Fig. 6b).

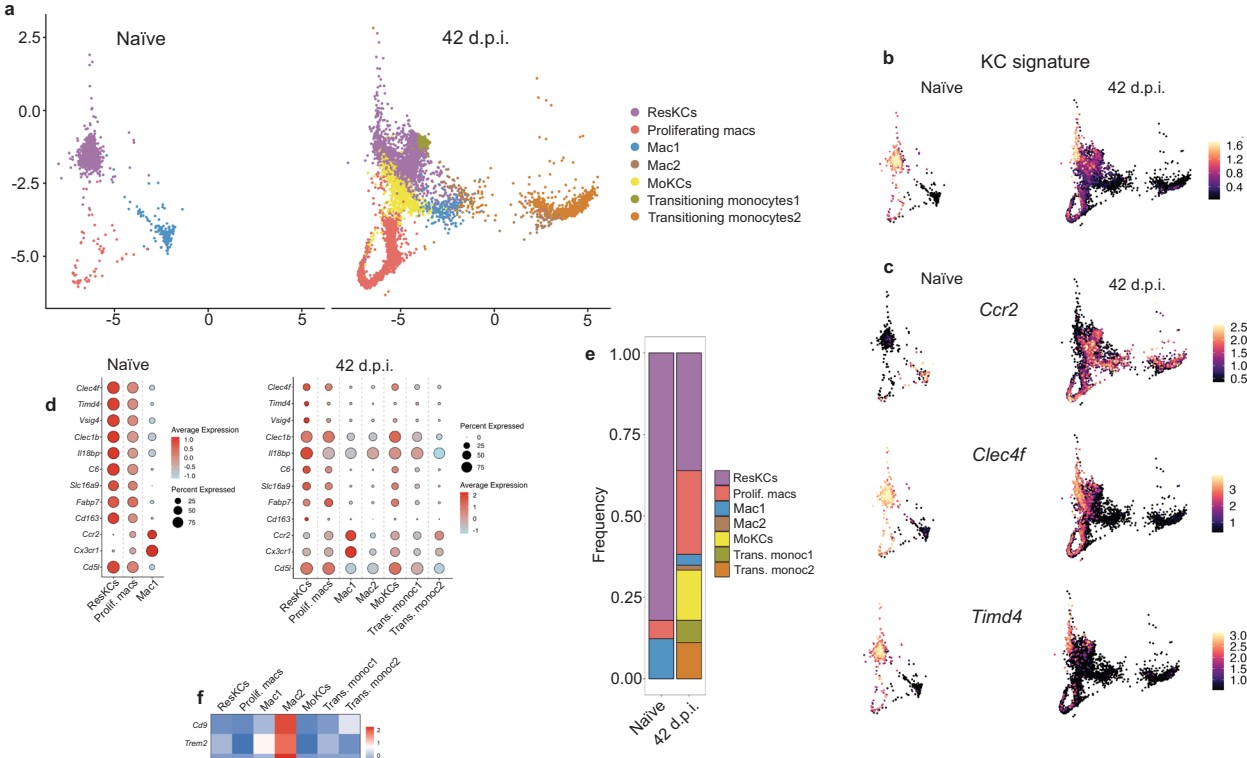

**Fig. 5 | VL-induced macrophage heterogeneity revealed by single-cell RNA sequencing. a** UMAP plot of scRNA-seq data from sorted live, single, CD45.2⁺F4/80⁺CD11b�^intCD64⁺ cells from livers of uninfected and 42 d.p.i. mice, showing three main clusters for naïve and seven clusters for infected mice. **b** Annotated UMAP plot showing the single-cell expression of conserved KC signature genes as described by Guilliams et al.[19] **c** Single-cell expression of *Ccr2, Clec4f*, and *Timd4* in CD45.2⁺F4/80⁺CD11bᵒ^intCD64⁺ cells from naïve and 42 d.p.i. mice. **d** Average gene expression and the percentage of cells expressing each gene within the identified clusters in naïve and 42-day infected CD45.2⁺F4/80⁺CD11bᵒ^intCD64⁺ cells. **e** Frequency of the different subsets identified by scRNA-seq in naïve and 42 d.p.i. mice. **f** Heatmap showing the differential expression of Mac2-specific genes identified by Remmerie et al.[14]. Data include 1200 naïve cells and 6152 cells from 42 d.p.i. mice after QC filtering.

Transitioning monocytes1 had the highest levels of pro-inflammatory chemokines and cytokines, including *Tnf, Il1b, Cxcl9, Cxcl10, Cxcl12* (Fig. 5b). Confocal microscopy revealed that iNOS expression peaked at 42 d.p.i. (Fig. 6c, d and Supplementary Fig. 7b) and was confined to CLEC4F⁻KCs and momacs within 42 d.p.i. granulomas (Fig. 6e, f). This suggests that *Nos2*-expressing cells in Transitioning monocytes1 and 2, and Mac2 clusters were likely localized within granulomas (Fig. 6g and Supplementary Fig. 7b). The highest expression of *Cxcl10* and *Tnf* was also detected in the same *Nos2* expressing clusters (Fig. 6g), reinforcing the pro-inflammatory nature of the granulomas. By contrast, *Il10* was detected mostly in cells from ResKCs and MoKCs clusters (Fig. 6g), and in CLEC4F⁻TIM-4⁺ sorted cells (Supplementary Fig. 7a).

Altogether, cells from Transitioning monocytes1 and 2 and Mac2 clusters expressed iNOS and high levels of pro-inflammatory cytokines and chemokines. Cells from the ResKCs cluster and CLEC4F⁻TIM-4⁺ sorted cells expressed some pro-inflammatory chemokines but also IL-10. MoKCs showed a more homeostatic and regulatory profile, expressing IL-10 and fewer chemokines and cytokines. Our data indicates significant functional heterogeneity among KCs and recruited macrophages, likely due to differences in ontogeny, length of time since engraftment, and spatial localization within the infected liver.

Previous studies using colloidal carbon or fluorescent nanobeads to label KCs before infection showed that *L. donovani* amastigotes were present in KCs within granulomas from 8-28 d.p.i[23,26]. Using confocal microscopy, *L. infantum* was found mostly inside CLEC4F⁺TIM-4⁺resKCs at 19 d.p.i. (Fig. 6h, i and Supplementary Fig. 7c). Forty-two d.p.i. was the peak of granuloma formation (Fig. 6j), and amastigotes were mostly found in CLEC4F⁻KCs within granulomas, with minimal presence in momacs (Fig. 6h, i, k and

Supplementary Fig. 7c). This finding suggests that resKCs are the primary infected cells during early infection, which become CLEC4F⁻KCs within granulomas. This raises the question as to what role the uninfected momacs are playing in the development of the anti-parasitic response in the liver.

Deficient monocyte recruitment during *L. donovani* infection has been previously shown to result in disorganized granulomas and increased parasite burdens[43–45]. Similarly, we found that *Ccr2*⁻/⁻ mice had 16.8-fold more liver parasites than WT mice at 42 d.p.i. (Fig. 6l). Quantification by confocal microscopy showed that resKCs and CLEC4F⁻KCs of the *Ccr2*⁻/⁻ mice harbored 4-fold more parasites than WT mice (Fig. 6m, n and Supplementary Fig. 7d). In the absence of recruited macrophages, CLEC4F⁻KCs and some resKCs from *Ccr2*⁻/⁻ mice both expressed iNOS (Supplementary Fig. 7e, f) but could not control the infection as effectively as WT mice. Liver homogenates from infected WT and *Ccr2*⁻/⁻ mice showed a different cytokine and chemokine environment (Supplementary Fig. 7g). Chemokines associated with granuloma formation, such as CCL2, CCL3 and CXCL10 were increased in *Ccr2*⁻/⁻ compared to WT, as was IFN-γ. However, disease-promoting cytokines such as IL-10 and IL-6[46] were also increased. The contribution of momacs to hepatic resistance is likely due to their cooperative functions with CLEC4F⁻KCs in the effector response. However, the increased proliferation of resKCs observed in infected *Ccr2*⁻/⁻ mice (Supplementary Fig. 2c) suggests that monocyte recruitment into the open niche might also serve to limit the homeostatic proliferation of resKCs, which better supports parasite survival and growth. In either case, a heterogeneous population containing both KCs and momacs is important for hepatic immunity in VL.

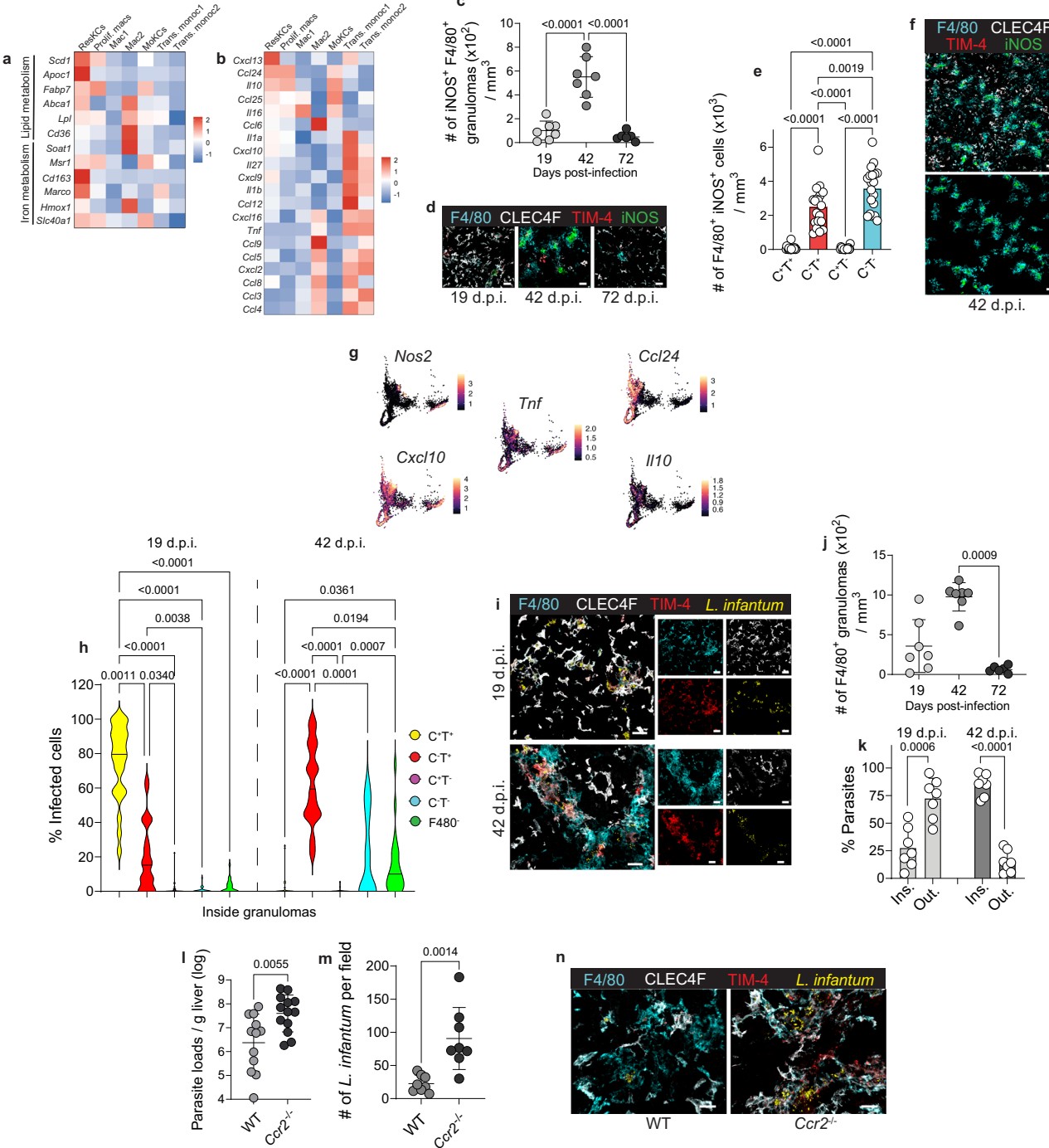

## Discussion

In this study, we highlight the importance of ontogenic macrophage heterogeneity in the control of VL. During late infection, KCs redistribution outside the sinusoids and ferroptosis were revealed as two potential mechanisms for creating an open KC niche, leading to its repopulation by moKCs within the sinusoids. Granuloma expansion in the perisinusoidal space was associated with remodeling of the sinusoidal network, and late-stage granulomas contained a heterogeneous macrophage population consisting of CLEC4F⁻KCs and momacs, both of which were iNOS⁺. ScRNA-seq revealed distinct functional profiles among these macrophage subsets. Momacs appeared highly proinflammatory, moKCs acquired regulatory functions, and CLEC4F⁻KCs displayed dual roles, involving the expression of both proinflammatory chemokines and IL-10. In infected *Ccr2⁻/⁻* mice,

impaired monocyte recruitment, increased resKCs proliferation, and reduced granuloma macrophage heterogeneity were associated with higher liver parasite burdens.

The signals influencing resident and monocyte-derived macrophages during infections are likely distinct from those in models of liver injury or targeted macrophage depletion and may also vary across different pathogens. For instance, *Listeria monocytogenes* infection triggered an early, nearly complete necroptotic death of KCs, followed by their replacement by momacs[9]. Similarly, murine models of viral hepatitis showed massive resKCs loss, which were rapidly replaced by monocyte-derived cells[11]. By contrast, our chronic VL model evidenced a more gradual and limited replacement of resKCs by moKCs and accumulation of momacs. Nevertheless, changes in the hepatic macrophage compartment during *L. infantum* infections were substantial

**Fig. 6 | Functional heterogeneity of macrophage subsets and the contribution of monocyte-derived macrophages to *L. infantum* control. a, b** Relative expression of lipid and iron metabolism (**a**), and chemokines and cytokines (**b**) genes from sorted CD45.2⁺F4/80⁺CD11bⁱⁿᵗCD64⁺ cells at 42 d.p.i. Data was down-sampled to a maximum of 500 cells per cluster. **c** Number of iNOS⁺F4/80⁺ granulomas in WT mice, quantified from immunofluorescence images. Data pooled from 2 independent experiments (*n* = 7 for 19- and 42 d.p.i. and *n* = 6 for 72 d.p.i.). **d** Representative images showing F4/80(cyan), CLEC4F(white), TIM-4(red), and iNOS(green) in WT livers. Scale bars, 30 μm. **e** Number of F4/80⁺iNOS⁺ macrophages in WT livers at 42 d.p.i. Data pooled from 5 independent experiments (*n* = 19). **f** Representative images showing F4/8(cyan), CLEC4F(white), TIM-4(red), and iNOS(green) (top), F4/80⁺iNOS⁺ granulomas (bottom) in WT liver at 42 d.p.i. Scale bars, 100 μm. **g** Single-cell expression of *Nos2*, chemokines, and cytokines in CD45.2⁺F4/80⁺CD11bⁱⁿᵗCD64⁺ cells at 42 d.p.i. **h** Frequency of infected cells within F4/80⁺ granulomas, obtained from 2-3 regions from each liver (ROIs=24 for 19 d.p.i., ROIs=23 for 42 d.p.i.). Data pooled from 2 independent experiments for each time point, 8 mice at 19- and 42 d.p.i., 7 mice at 72 d.p.i. **i** Representative images of F4/80⁺ macrophages in WT livers, stained with anti-F4/80(cyan), anti-CLEC4F(white), anti-TIM-4(red), and anti-*L. infantum*(yellow). Scale bars, 30 μm. **j** Number of F4/80⁺ granulomas quantified from immunofluorescence images. Data pooled from 2 independent experiments (*n* = 7 for 19- and 42 d.p.i., *n* = 6 for 72 d.p.i.). **k** Distribution of parasites inside and outside F4/80⁺ granulomas. Data pooled from 2 independent experiments (*n* = 7). **l** Parasite loads in WT and *Ccr2*⁻ mice at 42 d.p.i. Data pooled from 3 independent experiments (*n* = 12 for WT and *n* = 13 for *Ccr2*⁻/⁻). **m** Number of *L. infantum* amastigotes per field in 42-day infected WT and *Ccr2*⁻/⁻ mice, quantified from confocal microscopy images. Data pooled from 2 independent experiments (*n* = 8). **n** Representative images showing F4/80(cyan), CLEC4F(white), TIM-4(red), and *L. infantum*(yellow) in WT and *Ccr2*⁻/⁻ livers at 42 d.p.i. Scale bars, 20 μm. Values in (**c**, **e**, **h**, **j**–**m**) represent the mean ± SD. *P* values were obtained in (**c**, **e**) using ordinary one-way ANOVA with Tukey's multiple comparisons test, in (**h**, **j**) using Kruskal-Wallis test with Dunn's multiple comparisons tests, in (**k**–**m**) using a two-tailed unpaired *t* test. Source data are provided as a Source Data file.

over time, with over 70% of macrophages at 42 d.p.i. originating from monocytes.

In our model, TIM-4 served as a useful marker to differentiate embryonic vs monocyte-derived macrophages, while CLEC4F distinguished conventional resKCs from CLEC4F⁻KCs within granulomas outside the sinusoids. Importantly, our findings emphasize the need for caution when using CLEC4F as the primary KC marker, as its expression may be downregulated under certain conditions.

Spatial redistribution of KCs along the sinusoidal network has been observed in livers infected with *Mycobacterium bovis* BCG[47] and *L. donovani*[23]. During VL, KC redistribution has been demonstrated by the administration of nanobeads before infection as a method to track the KCs, which were observed to subsequently cluster at the core of the granulomas[23]. This suggested that KCs migrate to form granulomas. Using parabiotic and *Clec4f*^Cre-TdT^*ZsGreen* mice, we confirmed that KCs are found within granulomas and revealed that they downregulate CLEC4F in late-stage granulomas. By labeling the sinusoids in our confocal imaging, and further assessing vascular integrity by live imaging of labeled RBCs, the findings also suggest that the redistribution of KCs involves their migration into the perisinusoidal space, physically displacing the sinusoidal network but without causing any apparent vascular damage. The association of RBCs with KCs, possibly related to their clearance of dead or damaged erythrocytes, may explain the accumulation of some static RBCs within granulomas, carried there by KCs during granuloma formation. In addition, a previous study has demonstrated that KC depletion using clodronate can temporarily disrupt the sinusoidal layer, leading to structural gaps[48]. Similarly, the redistribution of KCs outside the sinusoids may contribute to the formation of such gaps, further accounting for the presence of RBCs in granulomas.

Under steady-state conditions, KCs span both the sinusoids and part of the perisinusoidal space, extending their processes to multiple sinusoids[6]. Our findings using the *Clec4f*^Cre-TdT^*Confetti* mice demonstrate that granulomas contain KCs from multiple clones, evidencing that KCs redistribution contributes to the expansion of the granuloma. Increased proliferation of momacs and CLEC4F⁻KCs was observed in *L. infantum* infected livers, suggesting that proliferation, along with the accumulation of other innate and adaptive cells around the KC core, also promotes granuloma expansion. This expansion led to physical displacement of the sinusoidal network, disrupting the contact between KCs within granulomas and circulating blood. In late-stage granulomas, KCs at the core downregulated CLEC4F and expressed iNOS and pro-inflammatory chemokines, but otherwise maintained a gene expression profile largely similar to that of CLEC4F⁺resKCs. These changes are possibly linked to the loss of sinusoidal contact and/or the pro-inflammatory environment within the granulomas. CLEC4F

downregulation has been reported in models targeting *Alk1* involved in maintaining KC homeostasis[49], and during liver fibrosis[50]. In these contexts, CLEC4F downregulation was associated with major changes in gene expression and a more comprehensive loss of KC identity.

KC redistribution to form granulomas may leave an open niche within the sinusoids. During BCG infection, Egen et al.[47] observed KC redistribution to form granulomas without changes in the overall density of F4/80⁺ cells within the sinusoids, implying that KCs were replaced by monocyte-derived cells recruited from the blood. Our findings support this idea, demonstrating that some F4/80⁺ cells inside the sinusoids were CLEC4F⁺moKCs, and suggesting that KC redistribution during granuloma formation triggers repopulation of the sinusoidal niche by moKCs.

Another possible explanation for the replacement of resKCs by moKCs during *L. infantum* infection involves resKCs death. Targeting pyroptotic and necroptotic cell death modalities during infection did not influence resKC accumulation. In contrast, apoptosis was unaffected in resKCs but increased in other F4/80⁺ macrophages. Furthermore, deletion of BACH1 to prevent ferroptosis[38,51,52] revealed that this transcription factor plays an important role in negatively regulating resKC and F4/80⁺ macrophages accumulation in *L. infantum* infected livers. Notably, in the infected *Bach1*⁻/⁻ mice, reduced lipid peroxidation was associated with increased resKCs proliferation compared to WT mice. This suggests that inhibiting the oxidative stress pathway promotes resKCs proliferation and increases their resistance to ferroptosis. Supporting these findings, a model of metabolic dysfunction-associated steatohepatitis (MASH) demonstrated that elevated iron content, associated with lipid peroxidation, led to KC ferroptotic cell death[53].

Altogether, these observations suggest that resKCs death, along with their redistribution in the perisinusoidal space to form granulomas, may contribute to an open sinusoidal niche and their partial replacement by moKCs within the sinusoids.

The scRNA-seq analyses and confocal staining provided clues as to the functionality of the different macrophage subsets during infection. CXCL13 and CCL24 are chemokines typically expressed by naïve resKCs[19] and resident macrophages in various tissues[54,55], respectively. Our scRNA-seq analysis revealed that both resKCs and CLEC4F⁻KCs maintained the expression of these chemokines. CXCL2 is known to mediate the chemotaxis of immune cells, CCL3 and CCL4 play roles in the development of Th1 cells, while CCL3, in combination with CCL2, enhances parasite killing of both *L. infantum* and *L. donovani*[56,57]. These chemokines were predominantly expressed by CLEC4F⁻KCs, which were also iNOS⁺ within granulomas, suggesting that KCs at the core of granulomas actively contribute to granuloma expansion and parasite control. Conversely, these cells exhibited higher levels of IL-10 compared to other macrophage subsets,

indicating their role in modulating excessive inflammation but potentially favoring parasite persistence. MoKCs displayed a gene expression profile similar to that of resKCs, with some cells expressing IL-10 but low levels of chemokine and cytokine expression. Since these cells appeared to repopulate the open KC niche within the sinusoids, it is reasonable to hypothesize that they would acquire the tissue-specific, homeostatic functions of resKCs. This aligns with findings from *L. monocytogenes* infections, where momacs replaced dead KCs following bacterial clearance and adopted a type 2 polarization to restore liver homeostasis[9].

In contrast to moKCs, momacs exhibited the highest expression of pro-inflammatory cytokines and chemokines. Given that the majority of macrophages within the 42-day granulomas were monocyte-derived, iNOS⁺, and expressed pro-inflammatory mediators, our findings strongly suggest that momacs are crucial for optimal granuloma function and parasite killing. Since *L. infantum* amastigotes were found predominantly within CLEC4F⁺KCs, the contribution of diffusible NO from uninfected momacs within the granuloma may be necessary to achieve an optimal NO concentration for killing. This may be important because individual iNOS-expressing cells were previously shown to be insufficient to control *Leishmania* growth[58]. The contribution of monocyte-derived cells to the anti-*Leishmania* response is supported by the increased parasite loads in the infected *Ccr2*⁻/⁻ mice. In these mice, the gradual reduction of resKC numbers within the sinusoids due to migration and death and the reduced monocyte infiltration, appear to be compensated by the enhanced proliferation of the remaining KCs. Additionally, M-CSF and IL-4, which are known to enhance tissue-resident macrophage proliferation during helminth infections[59], were elevated in *L. infantum* infected *Ccr2*⁻/⁻ livers compared to WT controls. It should be noted that the proliferation of KCs and the deficit of momacs in infected *Ccr2*⁻/⁻ mice did not prevent granuloma formation or iNOS expression by KCs within granulomas. Whether by preventing the compensatory expansion of KCs that better support parasite growth, and/or directly contributing to the killing response, the findings show the importance of momacs in parasite control during VL. Overall, our results highlight the critical role of the ontogenic heterogeneity of macrophages in shaping hepatic granuloma maturation and parasite control during VL.

## Methods

### Ethical statement

Our research complies with all relevant ethical regulations. All mice used in these studies were handled under a study protocol approved by the NIAID Animal Care and Use Committee (protocol number LPD 68E). All aspects of animal use in this research were monitored for compliance with The Animal Welfare Act, the PHS Policy, the U.S. Government Principles for the Utilization and Care of Vertebrate Animals Used in Testing, Research, and Training, and the NIH Guide for the Care and Use of Laboratory Animals.

### Mice

All mice used in this study were female, 6–8 weeks old, and housed at the NIAID animal care facility under specific pathogen-free conditions, at a constant cycle of 14 h in the light (< 300 lux) and 10 h in the dark. Colonies were maintained at 20–24 °C and 40–60% humidity, with free access to food and water. The following mouse strains were used: C57BL/6NTac (Taconic), B6 CD45.1 Jackson line #002014 (Taconic #8478), *Ccr2*⁻/⁻ Jackson line #004999 (Taconic #8456)[60], *Caspase1*⁻/⁻ (Taconic #8460)[61], *Clec4f-Cre-tdTomato* (JAX stock #033296)[12], *Ai6(RCL-ZsGreen)* (JAX stock #007906), *R26R-Confetti* (JAX stock #013731)[62], *Bach1*⁻/⁻ [63], *Mlkl*⁻/⁻ (JAX stock #037116)[64]. Strains #8478, #8456, and #8460 were obtained through a supply contract between the National Institute of Allergy and Infectious Diseases (NIAID) and Taconic Farms. Strains #033296, #007906, and #013731 were purchased from The Jackson Laboratory. *Bach1*⁻/⁻ [63] mice were generously

provided by Dr Kazuhiko Igarashi (Tohoku University Graduate School of Medicine). *Mlkl*⁻/⁻ mice were kindly provided by Dr. James Murphy (The Walter and Eliza Hall Institute of Medical Research). *Clec4f^{Cre-TdT}ZsGreen* mice were obtained by breeding *Clec4f-Cre-tdTomato* x *Ai6(RCL-ZsGreen)* mice, and *Clec4f^{Cre-Tdt}Confetti* mice were obtained by breeding *Clec4f-Cre-tdTomato* x *R26R-Confetti* mice.

### Parasites

*Leishmania infantum* (MHOM/ES/92/LLM-320; isoenzyme typed MON-1) was cultured in M199 medium supplemented with 20% fetal bovine serum (Gemini Bio-Products), 100 µ/mL penicillin and 100 µg/mL streptomycin (Gibco), 2 mM L-glutamine (Gibco), 40 mM HEPEs (Gibco), 0,1 mM adenine (Sigma) in 50 mM HEPEs, 5 mg/mL hemin (Sigma) in 50% triethanolamine (Sigma), and 1 mg/mL 6-biotin (Sigma). Cultures were maintained at 26 °C. For mouse infections, metacyclic promastigotes were purified from stationary-phase cultures by Ficoll (Sigma) density gradient (8% and 20%), as adapted from Spath and Beverley[65], and centrifuged at 500 × *g* for 10 min, at 25 °C. Mice were intravenously inoculated with 3 × 10⁶ metacyclic promastigotes in 100 µl of 1 × PBS.

### Leukocytes isolation and purification from the liver

Mice were euthanized in a CO₂ chamber and immediately perfused with 20 mL sterile 1x PBS (Gibco). The gallbladder was removed, and the livers were weighed and maintained in 1 × PBS at 4 °C until processing. Livers were place in 3 mL of a digestion solution containing 0.5% collagenase IV (Worthington) and 0.5 mg/mL DNAse (Sigma) in RPMI medium (Gibco) and incubated for 30 min at 37 °C. The digested livers were filtered through a 70 µm cell strainer using 1 × PBS. For leukocyte purification, cells were centrifuged in a 34% Percoll (GE Healthcare) solution, prepared in 1 × PBS, at 500 × *g* for 20 min at 25 °C. Red blood cells were lysed using ACK lysis buffer (Lonza), and the remaining cells were washed once with 1x PBS (Gibco).

### Immunolabeling for flow cytometry

All cells were stained using the LIVE/DEAD Fixable Blue Dead Cell Stain Kit (Thermo Fisher Scientific) at a 1:1000 dilution in 1 × PBS, for 20 minutes at 4 °C in the dark. Nonspecific labeling was blocked with TruStain FcX™ (anti-mouse CD16/32) antibody (Biolegend) at a 1:100 dilution. Surface markers used include the following: CD45.1 (A20, BD Biosciences), at 1:200, CD45.2 (104, BD Biosciences) at 1:200, F4/80 (BM8, Biolegend) at 1:200, CD11b (M1/70, Biolegend) at 1:800, CD11c (N418, Biolegend) at 1:200, Ly6C (HK1.4, Biolegend) at 1:400, Ly6G (1A8, Biolegend) at 1:200, Siglec F (S17007L, Biolegend) at 1:200, MHCII (M5/114.15.2, Biolegend) at 1:800, CLEC4F (3E3F9, Biolegend) at 1:50, CD64 (X54-5/7.1, Biolegend) at 1:200, TIM-4 (RMT4-54, eBioscience™) at 1:200. All antibodies were diluted in FACS buffer (0.5% FBS + 1 mM EDTA in 1 × PBS). Samples were washed once with FACS buffer, fixed, and permeabilized with BD Cytofix/Cytoperm™ (BD Biosciences) for 30 min at 4 °C in the dark. Cells were washed once with FACS buffer and resuspended in FACS buffer mixed with Accu-Check Counting Beads (Thermo Fisher Scientific). Labeled leukocytes were acquired using the Cytek® Aurora cytometer (Cytek® Biosciences), and analysis was performed with SpectroFlo® software (Cytek® Biosciences, version 3.0.1 to 3.3.0) and FlowJo™ software (Treestar®, version 10.8.1 to 10.10.0).

### Quantification of parasite loads

Livers were perfused with 20 mL of 1 × PBS, and leukocytes isolated as described above. The cells were resuspended in 1.5–2.5 mL of 1 × PBS. Parasite loads were quantified similarly to Lee et al.[55]. Briefly, a volume of 50–80 µL was used for 2-fold serial dilutions in a 96-well-flat bottom plate containing 150 µL of M199/S medium per well, incubated at 26 °C. After 7–10 days, parasite presence was evaluated, and the

parasite load per gram of tissue was calculated based on the highest dilution where *L. infantum* was detected.

## Parabiosis

CD45.1$^+$ and CD45.2$^+$ mice with similar weight were cohoused for 2 weeks before undergoing surgical connection as described previously[66]. Three weeks post-surgery, both mice in the pair were infected and euthanized 42 days post-infection.

## Intravital microscopy

All intravital imaging experiments were approved by and conducted in accordance with the guidelines of the Animal Care and Use Committee of the National Institute of Allergy and Infectious Diseases. Intravital imaging of the liver was performed as previously described[67]. Briefly, images were acquired using a Leica DIVE (Deep In Vivo Explorer) inverted microscope (Leica Microsystems), equipped with a full range of visible light lasers (Spectra-Physics) and a 37 °C environmental chamber (NIH Division of Scientific Equipment and Instrumentation Services). Anesthesia was provided using a SurgiVet vaporizer and a nose-cone mask (Braintree Scientific). Infected *Clec4f*$^{Cre-TdT}$*ZsGreen* or *Clec4f*$^{Cre-Tdt}$*Confetti* mice were anesthetized with 2% isoflurane delivered into the induction chamber (Braintree Scientific) and maintained at 1.75% during imaging. The imaging was performed in regular confocal mode using a long-working distance objective (L HC FLUOTAR 25x, NA = 0.95, Leica Microsystems). Antibodies were injected intravenously and consisted of the following: anti-CD31 AF647 or BV421 (clone MEC13.3, Biolegend) to visualize liver blood vessels, F4/80 BV421 of AF488 (clone BM8, Biolegend), or Ter119 PE (clone TER-119, Biolegend). The liver was surgically exposed, stabilized using a custom-built tissue holder, and immersed in pre-warmed lubricating jelly. Static images were tiled, and a merged mosaic was constructed using Leica tile scanning software. For image quantification, selected cells were rendered as 3D surface objects using Imaris software (Andor, version 9.7.2 to 10.2.0). Vein reconstruction and RBC autoregressive track analysis was performed in Imaris utilizing the spot function. Channel intensity statistics were exported as ims files, converted to CSV datasets using a Python parser, and imported into FlowJo™ (Treestar®, version 10.8.1 to 10.10.0) for gating and analysis. Image analysis was conducted with Imaris software (Andor, version 9.7.2 to 10.2.0). The parser was written in Python, and the source code is available at https://zenodo.org/records/14970154[68].

## Anti-Leishmania antibody purification from serum

To visualize *L. infantum* amastigotes by confocal microscopy, we used pooled serum samples from kala-azar cured patients[69]. Samples were centrifuged at 10,000 × *g* for 10 minutes at 4 °C, and IgG purified using the Pierce™ Protein A IgG Purification Kit (Thermo Fisher Scientific) following the manufacturer's instructions. The eluted antibodies were pooled, dialyzed in 1x PBS (Three 2-hour exchanges and one overnight exchange, at 4 °C), and concentrated using the Amicon® Ultra-4 50 K (Merck Millipore) at maximum speed for 15 min. The antibodies were lyophilized and stored at − 80 °C until use. Before staining, the antibodies were resuspended in 1 × PBS, conjugated with the ReadiLink™ xtra Rapid iFluor® 488 Antibody Labeling Kit (AAT Bioquest), and used for staining fixed/frozen liver sections.

## Confocal microscopy

The fixation and staining protocol was adapted from Radtke et al.[70]. Briefly, livers were perfused with 1 × PBS, and a piece of the larger lobule was fixed with BD Cytofix/Cytoperm™ (BD Biosciences) diluted 1:4 in 1 × PBS for 24 h at 4 °C. The tissues were then immersed in 30% sucrose (Sigma) solution overnight at 4 °C, placed on a Histomold (Fisher Healthcare), and embedded in Tissue-Tek® O.C.T. Compound (Sakura Finetek). The histomold was frozen on dry ice and stored at − 80 °C. OCT-embedded tissues were sectioned using a cryostat (Leica

Biosystems), and 10 µm sections were prepared on VWR® Superfrost® Plus Micro slides (VWR Corporate Headquarters). The samples were kept frozen at − 80 °C until use. For immunostaining, tissue sections were rehydrated with 1x PBS for 5 min at RT and blocked with a buffer containing 1% TruStain FcX™ (Biolegend) and 0.3% Triton X-100 (Calbiochem®), in 1% BSA (Thermo Fisher Scientific) for 1 hour at 37 °C. The samples were incubated with the following antibodies when needed: F4/80 (BM8, eBioscience™) at 1:100, CLEC4F (3E3F9, Biolegend) at 1:100, TIM-4 (RMT4-54, Biolegend) at 1:200, iNOS (CXNFT, eBioscience™) at 1:200, CD138 (281-2, Biolegend) at 1:300, Cleaved Caspase-3 (Asp175) (Polyclonal, Cell Signaling Technology) at 1:100, Ki-67 (B3B5, Cell Signaling Technology) at 1:100, and/or anti-*L. infantum* (purified and conjugated in-house) at 1:200. The antibodies were diluted in 1% BSA containing 0.3% Triton X-100 and 1:5000 Hoechst 33342 (Thermo Fisher Scientific) for 1 h at 37 °C. After incubation, the sections were washed 5 times for 3 minutes each with 1 × PBS. Slides were mounted using Prolong™ Diamond Antifade Mountant (Thermo Fisher Scientific), following the manufacturer's instructions. Tiled images (3 × 4 for naïve and 4 × 4 for infected mice) were acquired using the Leica SP8 WLL FLIM confocal microscope. Tiles were merged using LAS X Navigator software (LAS X 3.5.5.19976) and analyzed with Imaris software (Andor, version 9.7.2 to 10.2.0). Image quantification was performed as described above.

## Lipid peroxidation assay and intracellular GSH measurement

Lipid peroxidation in F4/80$^{hi}$CD11b$^{int}$CD64$^+$TIM-4$^{+/−}$ and F4/80$^{low/−}$CD11b$^+$ cells was assessed using BODIPY™ 665/676 (Lipid Peroxidation Sensor) (Thermo Fisher Scientific). Briefly, cells were isolated as described above and incubated with 10 µM Bodipy in 1 × PBS for 30 min at 37 °C. After incubation, cells were washed once with 1 × PBS, and extracellular staining was performed as described above.

Glutathione (GSH) levels were measured in liver homogenates. Livers were perfused with 20 mL of 1 × PBS, and a piece of the larger lobule was homogenized in 1 × PBS using glass beads. The homogenate was centrifuged at maximum speed at 4 °C for 10 min to remove tissue matrix and cell debris. Supernatants were stored at − 80 °C until use. Reduced glutathione levels were measured using the Glutathione Assay Kit (Cayman, USA, 703002) following the manufacturer's instructions. Total protein was determined using the Pierce Protein Assay (Thermo Fisher Scientific), according to the manufacturer's instructions, and values used to normalize GSH levels based on total protein content.

## Single-cell RNA sequencing

Liver leukocytes were isolated as described above, and tissue digestion/Percoll solution and staining buffers were supplemented with 100 µ/mL SUPERase•In™ RNase Inhibitor (Thermo Fisher Scientific). Live, single CD45.2$^+$ F4/80$^{hi}$ CD11$^{int}$ CD64$^+$ macrophages or CD45.2$^+$ F4/80$^{hi}$ CD11$^{int}$ CD64$^+$CLEC4F$^-$TIM-4$^+$ KCs were sorted using the Cytek® Aurora CS sorter and collected for RNA extraction and library preparation using the Chromium Next GEM Single Cell 3′ Kit v3.1 (PN-1000269, 10x Genomics), according to the manufacturer's instructions. In brief, single cells were encapsulated in aqueous droplets containing beads coated with barcoded oligo-dT in a water-in-oil emulsion using a Chromium controller (10x Genomics). RNA transcripts from lysed cells were reverse-transcribed into cDNA, followed by amplification, fragmentation, dual-adapter ligation, and size selection. Libraries were analyzed using a D5000 ScreenTape on a TapeStation (Agilent) and sequenced on an Illumina HiSeqX Ten by Psomagen Inc.

Sequence reads were demultiplexed into FASTQ files from raw base calls using the *mkfastq* module in Cell Ranger v.5.0.0 (10x Genomics). Quality control of the raw FASTQ files was performed using FastQC, and reads with an average quality score of < 20 or a length of < 20 bp were filtered out using fastp v.0.20[71]. Cell Ranger v.7.1.0 *count*

was used to align filtered reads to the mouse reference transcriptome mm10 (GENCODE vM23/Ensemble 98) and to quantify the number of different unique molecular identifiers (UMIs) for each gene. For F4/80[hi]CD11b[int]CD64[+] sorted cells, the proportion of reads with valid barcodes assigned to cells and confidently mapped to the transcriptome ranged from 45.3–60.6% in uninfected samples and 86.7–89.3% in infected samples. For F4/80[hi]CD11b[int]CD64[+]CLEC4F[-]TIM-4[+] sorted cells, 78.8% of the barcodes were valid. The estimated number of F4/80[hi]CD11b[int]CD64[+] cells detected per replicate ranged from 1166 and 4804 cells, totaling 11,834 cells. For F4/80[hi]CD11b[int]CD64[+]CLEC4F[-]TIM-4[+] cells, the total was 421 cells. These data were used as input for downstream processing and analysis using Seurat v.4.3.0[72] in RStudio version 2023.03.0.

Cells were excluded from analysis if mitochondrial gene expression exceeded 10% of total expression, expressed less than 1000 genes, more than 6000 genes, or fewer than 5000 transcript molecules were detected. Genes expressed in fewer than 2 cells were also removed. UMI counts were normalized by library read depth, log-transformed, centered, and Z-scored using functions *NormalizeData* and *ScaleData*. Highly variable genes were identified using *FindVariableFeatures* (2000 features, "vst"method). Data integration across samples was performed using *SelectIntegrationFeatures* to identify variable genes, *FindIntegrationAnchors* to identify anchor cells, and *IntegrateData* to merge datasets. Dimensionality reduction was conducted on scaled integrated data using *RunPCA* and *RunUMAP* and clustering with *FindNeighbors* and *FindClusters* (15 principal components and a resolution of 0.2). To be consistent with annotation from the literature on KCs, we mapped our data onto a reference scRNA-seq dataset from Remmerie et al.[14]. Anchor point cells were selected between the reference and our dataset query using the *FindTransferAnchors* function, which is based on shared gene expression and mutual nearest neighbors (MNNs) in a shared low-dimensional space. Next, we used those anchors to assign cell type labels and project the reference dimensional reduction onto our dataset using the *MapQuery* wrapper. Cluster-specific marker genes were identified using *FindAllMarkers* and evaluating the average expression and the percentage expressed of each gene in a cluster compared to others. Markers were selected based on log2FC > 0.25 and statistical significance (adjusted *p*-value < 0.05, Wilcoxon rank-sum test). Clusters were manually annotated based on alignment with the reference dataset. A KC signature score was calculated using the *AddModuleScore* function with 15 human-mouse conserved KC markers[19] (*Cd5l, Vsig4, Slc1a3, Cd163, Folr2, Timd4, Marco, Gfra2, Adrb1, Tmem26, Slc40a1, Hmox1, Slc16a9, Vcam1, Sucnr1*).

**Multiplex cytokine and chemokine arrays**
Inflammatory mediators in liver homogenates were evaluated using approximately 45 mg of liver tissue. Tissues were homogenized in 200 μL of RIPA lysis buffer (Millipore) supplemented with protease inhibitor cocktail (Thermo Fisher Scientific) using Precellys tubes with metal beads (2 cycles of 20 s at 5000 rpm each). Homogenates were centrifuged at $8000 \times g$ for 10 min at 4 °C, and the supernatant was used for downstream analysis. Cytokines and chemokines were quantified using the commercial MILLIPLEX MAP mouse cytokine/chemokine magnetic bead panel kit (Millipore Sigma), according to the manufacturer's instructions. Cytokines and chemokines concentrations in pg/mL were normalized to liver tissue weight (mg).

**Statistical analyses**
Statistical analyses were conducted using GraphPad Prism software, version 10.0.0 to 10.2.0 (La Jolla). Data are presented as mean ± standard deviation (SD), with statistical significance defined as $p < 0.05$. The normal distribution and the homogeneous variance were tested for all analyses. For datasets following a normal distribution, unpaired Student's *t* tests (two-tailed) or one-way ANOVA with Tukey's

or Sidak's multiple comparisons were applied. For datasets not meeting normality assumptions, unpaired Mann-Whitney U tests (two-tailed) and Kruskall-Wallis tests with Dunn's post hoc analysis were used.

**Reporting summary**
Further information on research design is available in the Nature Portfolio Reporting Summary linked to this article.

## Data availability
The raw single-cell RNA-seq data generated in this study have been deposited in the NCBI SRA database under the BioProject ID PRJNA994082, available at https://www.ncbi.nlm.nih.gov/bioproject/PRJNA994082. The processed Seurat objects are available at https://doi.org/10.5281/zenodo.14246232[73]. Source data are provided in this paper.

## Code availability
The code is available at https://doi.org/10.5281/zenodo.14970154[68] and https://zenodo.org/records/14246232[73].

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

## Acknowledgements

We thank O. Schwartz and M. Smelkinson (Biological Imaging Section) for their assistance with microscope setup and image acquisition; Jarina DaMata, C. Eigsti, T. Moyer, and I. Douagi (Flow Cytometry Section) for their support with the cell sorting; P. Duncker (Cytek Biosciences) for guidance in designing the flow cytometry panel; and P'ng Loke for critically reviewing the manuscript. This work was partially supported by the Intramural Research Program of the NIAID, National Institutes of Health.

## Author contributions

Conceptualization, methodology, and validation, G.P., A.P., E.P.A., P.H.G.G., S.G., S.H.L., and D.L.S.; Investigation, G.P., A.P., E.P.A., P.H.G.G., and O.K.; Software, T.R.F., J.K., and S.R.P.; Data curation, T.R.F.; Formal analysis, G.P., T.R.F., and J.K.; Resources, D.L.S.; Writing - original draft, visualization, supervision, and project administration G.P. and D.L.S.; Writing – Review and Editing, G.P. and D.L.S.; Funding acquisition, D.L.S.

## Funding

## Competing interests

The authors declare no competing interests.
