## [Transparent Peer Review file · Nature Communications]

Kupffer cell and recruited macrophage heterogeneity orchestrate granuloma maturation and hepatic immunity in visceral leishmaniasis

Corresponding Author: Dr David Sacks

Version 0:

Reviewer comments:

Reviewer #1

(Remarks to the Author)

This manuscript by Pessendra et al., investigates hepatic macrophage heterogeneity in the context of visceral leishmaniasis. They report that upon infection, some resKCs become infected, migrate from the sinusoids to form granulomas where they lose CLEC4F expression. These resKC-derived macrophages (exKCs) are then complemented by monocyte-derived macrophages in the granulomas while the resKC pool in the sinusoids are repopulated through moKC development.

Understanding hepatic macrophage heterogeneity in different settings has become a key goal of many labs over recent years, thus the presence of distinct populations of macrophages in this context is not surprising or indeed that novel. However, there are a couple of more novel elements to this manuscript, for example the idea that ResKCs would migrate out of the sinusoids and subsequently dedifferentiate, which make it more interesting, although the data leading to these conclusions are not yet, in my opinion, 100% convincing.

Specifically, I have the following major and minor concerns with the manuscript and it's claims in the current form that would need to be addressed prior to publication.

Major Concerns:

1. Regarding the claim that KCs are de-differentiating or losing their identity outside of the sinusoids (in the granulomas). For now, this is claimed solely on the basis of CLEC4F expression. This is not enough to say these are ex-KCs. Can the authors show that these ex-KCs also lack expression of other KC markers? This could be done by flow cytometry e.g. by checking VSIG4, CD163 (intracellular), Fcrl2, Clec2 expression or more in-depth using qPCR and including a range of KC identity genes as used to define KCs in the single cell analysis in this manuscript (signature from Guilliams et al. Cell 2022).
2. Indeed, in this regard, in their scRNA-seq analysis the authors could not convincingly identify these exKCs, suggesting they either did not isolate them well or that they were perhaps in the transitioning monocyte fraction. Given that the authors find CLEC4F-TIM4+ cells in their flow analysis (extended data Fig 1) it seems unlikely that they are not isolating these cells for the scRNA-seq analysis. Additionally, the idea that a KC would lose its identity and regain monocyte markers seems a bit far-fetched. Overall, the authors need to better assess where these cells lie in their scRNA-seq analysis. This could be done in multiple ways. Given that the cells are there in their flow cytometry data, the authors could do CITE-seq analysis using CLEC4F and TIM4 barcoded antibodies to then find back the cells and assess their transcriptional profile. Perhaps even easier, the authors could just sort the CLEC4F-TIM4+ cells from the liver and run them by either scRNA-seq or even bulk RNA-seq and then overlay the signatures on their current data.
3. The annotation of the scRNA-seq analysis is based on published data from a NAFLD model, exactly how the annotations were transferred here is not clear, for example if MAC1 do not express Clec1b how similar are they really to the published MAC1s? What other elements of the signature do they share? Here a table with DEGs per cluster would have been very useful for comparison. The same also holds for the Trans mono 1 cluster.
4. The authors claim that the KCs migrate out of the sinusoids to form granulomas, but this claim is not substantiated. Could

it not equally be that these granulomas form in place and damage the sinusoids, leading to their loss? Indeed, the areas where they are present show no sinusoidal vessels but large areas like this do not exist in healthy liver. I am convinced that the granulomas are not in sinusoids at this timepoint, but the idea that the KCs are really migrating that is not conclusively shown. To really claim migration, the authors would need to demonstrate KC migration with intravital imaging or a confocal timecourse. Perhaps looking at an earlier timepoint (or a few) after infection would help here.

5. While the data is convincing that the CLEC4F-TIM4⁺ exKCs and CLEC4F-TIM4⁺ KCS are resident and the other macrophage populations monocyte derived based on the data provided, I find the CCR2KO data somewhat confusing and requiring further explanation. It is well known that even in the absence of CCR2 some monocytes/monocyte precursors can egress the BM and thus it has previously been shown that in a KC depletion model CCR2KO mice still develop moKCs but with a slightly slower kinetics as the few monocytes need to proliferate more to fill the niche (Bonnardel et al., Immunity 2019). Thus, here I question why there are suddenly no moKCs at all? One possibility is that the remaining ResKCs kick into proliferation in absence of monocytes to refill the niche as was shown in a partial KC depletion model (Scott et al., Nat Comms 2016). But this would suggest that niche repopulation in steady state also occurs via the two mechanisms proliferation and monocyte recruitment. Thus, the authors should check this in their model. Do resKCs proliferate (scRNA-seq data would suggest yes), and is this proliferation enhanced in CCR2KO mice? Do the exKCs also proliferate in the granulomas? With this in mind, could it be the proliferation rather than the loss of monocytes contributing to increased parasite load in CCR2KO mice? Here I think the claims should be toned down as with so much changing in the CCR2KO mice, I don't think the authors can be sure that this is due to lack of monocytes?

6. The authors also claim that resKCs die by ferroptosis leading to their replacement by moKCs. This is shown using Bach1KO mice as Bach1 deficiency is linked with decreased ferroptosis. However, again as this is a full KO mouse, I am not sure this is the only explanation for the data. Could it not also be that Bach1 deficiency interferes with monocyte recruitment/differentiation? How do monocytes look in the livers of these mice, are they still recruited? Do ResKCs proliferate more in this model? Could that explain the reduced need for moKC differentiation (see pt 5 above). After checking these aspects, it would also be interesting for the authors to look earlier post infection. Now they check D42 pi when moKCs are present thus the niche is already refilled (at least partially) perhaps checking earlier would allow them to better visualize the dying KCs? Although I certainly acknowledge that capturing cell death in vivo is not straight forward.

Minor Concerns:

1. The gating strategy in extended figure one, likely misses many macrophages (monocyte derived) that are not Cd11bint. Can the authors also show these gates in the infected mice and specify for the scRNA-seq analysis exactly what was sorted?
2. Why are there no capsule macrophages annotated in the scRNA-seq data?

(Remarks on code availability)

Reviewer #2

(Remarks to the Author)

In their manuscript „Kupffer cell and recruited macrophage heterogeneity orchestrate granuloma maturation and hepatic immunity in visceral leishmaniasis“, Pessenda et al. address the question of how the myeloid network in the liver is interacting and replenished in the context of *L. infantum* infection. Using a series of very elegant reporter system approaches which address the dynamics of Kupffer cells and monocyte-derived cells, the authors show evidence that during visceral Leishmaniasis, the resident Kupffer cell (resKC) niche is emptied through detachment from the liver sinusoids and relocalization to granulomas, as well as through ferroptotic cell death. In particular, the authors claim that resKC lose CLEC4F expression upon detachment from Liver sinusoidal endothelial cells (LSECs) and aggregation within granulomas, and, together with transitioning monocytes, exhibit an inflammatory expression profile, while resKC as well as monocyte-derived KC outside the granuloma show a homeostatic or immunoregulatory path. By employing *Ccr2*-deficient animals, the authors finally provide results that underline the importance of recruited monocyte for pathogen control.

Overall, this is a very interesting study which deserves attention from both infection immunologists as well as from the myeloid cell field. The experiments are beautifully designed for the scientific questions investigated, and the results fully support the conclusions drawn by the authors. While the characterization of the different KC and monocyte-derived cell turnover, population dynamics and localization, is admirable, the manuscript could be strengthened if some of the mechanistic aspects that the authors touch upon in the second part of the manuscript would be addressed more thoroughly (see below).

Major comments:

1. In Figure 3, the authors show evidence for the detachment of T-C⁺ KC from LSECs and hypothesize in accordance with Bonnardel et al, (Immunity 2019) that the lack of contact between KCs and LSECs might be responsible for their loss of identity, i.e. CLEC4F downregulation. In the said reference, DLL-Notch signaling is shown to be a main determinant of KC identity. Is the stimulus from the LSECs indeed continuously required, i.e. would blocking antibodies against DLL1 and DLL4 in the *L. infantum* model also result in enhanced identity loss of the KCs?

2. In their scRNAseq experiment, the authors identify inflammatory, granuloma-associated macrophages via the expression of iNOS, which they have shown using confocal imaging to be limited to granulomas. Also, they note that they could not identify a defined cluster of CLEC4F-Tim-4+. Could Leishmania infection, which the authors show nicely in their confocal microscopy, be used to characterize the infection rates in C+T+, C+T-, C-T+ and C-T- macrophages by flow cytometry to underline the observations from histology? Related to this, could the L. infantum RNA sequencing reads (which should be available from the 10x data) be mapped to the UMAP plots to characterize and assign infected cells to the different populations according to the L. infantum infection rate?

3. The authors use Ccr2-deficient mice which lack monocyte recruitment to demonstrate the importance of their findings for control of L. infantum and show that while the resKC still express iNOS under these conditions, this seems not to be sufficient to contain the infection. Moreover, IL-4 in the liver seems to be increased, which the authors conclude could additionally impair pathogen control in Ccr2-deficient mice. Could anti-IL-4 treatment at least partially rescue the phenotype? Alternatively, could a mixed Ccr2-DTR : iNOS^{-/-} bone marrow chimera approach be used to enable only the recruitment of iNOS^{-/-} monocytes, thus dissecting the influence of resKC-produced from monocyte-produced nitric oxide?

Minor points:

Figures 2J, 2K and 3c: The tdTomato fluorescence is hardly recognizable, possibly due to the resolution. This might be improved by showing the single channel images not for the whole images, but (also) for the magnification insets.

Line 117 use past tense: "the few that we detect"

Line 229 sentence hard to understand: "the Nos2 annotated UMAPs will have identified macrophages"

(Remarks on code availability)

Version 1:

Reviewer comments:

Reviewer #2

(Remarks to the Author)

The authors have responded to my major comments with additional experiments, and have appropriately addressed the points raised. I have no further comments and would like to congratulate the authors to this nice work.

(Remarks on code availability)

I lack the expertise to appropriately review the code provided by the authors

Reviewer #3

(Remarks to the Author)

Reviewer #1's comments centered on the dedifferentiation and migration of Kupffer cells, as well as the significance of monocyte recruitment. The authors have resolved the issues raised by the reviewer through additional experimental data, re-analysis, and reinterpretation of the results. The authors' responses and supplementary experiments have significantly bolstered the scientific robustness and persuasiveness of the article. Overall, I believe this manuscript is suitable for publication in Nature Communications.

(Remarks on code availability)

RESPONSE TO REVIEWER'S COMMENTS

We sincerely thank the reviewers for their valuable comments, suggestions, and insights, which have significantly improved our manuscript.

In the revised version, we have incorporated new data in Figure 3 and in Supplementary Figures 2, 3, 5, 6, and 7 in response to the reviewers' suggestions. Additionally, we have updated Supplementary Figure 1. The revision also includes 7 live imaging videos from both our previous and recent experiments, as well as two tables showing differentially expressed genes (DEGs) identified in our prior and newly scRNA-seq analysis.

Following the journal's guidelines, we have also improved the introduction and added a discussion section. All revisions are highlighted in yellow. The figures in this file address specific comments but have not been added to the final manuscript.

Reviewer #1 (Remarks to the Author):

This manuscript by Pessendra et al., investigates hepatic macrophage heterogeneity in the context of visceral leishmaniasis. They report that upon infection, some resKCs become infected, migrate from the sinusoids to form granulomas where they lose CLEC4F expression. These resKC-derived macrophages (exKcs) are then complemented by monocyte-derived macrophages in the granulomas while the resKC pool in the sinusoids are repopulated through moKC development.

Understanding hepatic macrophage heterogeneity in different settings has become a key goal of many labs over recent years, thus the presence of distinct populations of macrophages in this context is not surprising or indeed that novel. However, there are a couple of more novel elements to this manuscript, for example the idea that ResKCs would migrate out of the sinusoids and subsequently dedifferentiate, which make it more interesting, although the data leading to these conclusions are not yet, in my opinion, 100% convincing.

We agree that understanding macrophage heterogeneity is a goal of many labs, yet most research focuses on models of complete resKC depletion and/or non-infectious models of resKC replacement. The dynamics of an open KC niche in the context of chronic infection—where gradual and incomplete replacement of resKCs may occur—have not been carefully investigated. In addition, most studies emphasize KC death as the main mechanism for replacement by monocyte-derived cells. Here, we provide evidence that a potential cause of an open sinusoidal KC niche is their redistribution outside of the sinusoids to form granulomas.

The ontogenic diversity of macrophages has not been clearly defined in experimental VL, as monocyte-derived cells engrafting the KC niche can adopt KC-specific markers. Additionally, as we demonstrate with CLEC4F, these markers may be lost. Using CLEC4F and TIM-4 in addition to F4/80 to phenotype these cells at different stages of granuloma formation, our study provides a comprehensive characterization of macrophage and KC heterogeneity during VL. By integrating these phenotypic markers with data from *in situ* imaging, parabiotic and reporter mice, along with single-cell transcriptomics, we distinguish between embryonic and monocyte-derived macrophages, map their spatial distribution within granulomas and in relation to the sinusoidal network, and show their activation states and potential roles during infection. Thus, while we agree that KCs redistribution outside the sinusoids and CLEC4F downregulation are important aspects of our findings, our insights into macrophage composition, and the contribution of this heterogeneity to granuloma maturation and hepatic resistance, are substantial advancements in the study of experimental VL.

Specifically, I have the following major and minor concerns with the manuscript and its claims in the current form that would need to be addressed prior to publication.

Major Concerns:

1. Regarding the claim that KCs are de-differentiating or losing their identity outside of the sinusoids (in the granulomas). For now, this is claimed solely on the basis of CLEC4F expression. This is not enough to say these are ex-KCs. Can the authors show that these ex-KCs also lack expression of other KC markers? This could be done by flow cytometry e.g. by checking VSIG4, CD163 (intracellular), Folr2, Clec2 expression or more in-depth using qPCR and including a range of KC identity genes as used to define KCs in the single cell analysis in this manuscript (signature from Guilliams et al. Cell 2022).

In our initial description of the C-T⁺ cells, we did not intend to suggest that these cells were resKCs undergoing de-differentiation and acquiring monocyte markers. Rather, we aimed to convey that these cells had lost the most specific KC marker - CLEC4F - and appeared to express some pro-inflammatory markers, as many were iNOS⁺ in our confocal imaging. This could have led to their clustering with transitioning monocytes in the scRNA-seq analysis. Additionally, our terminology and the description suggesting they were "losing their identity" might have implied that these cells were ex-KCs. To reduce confusion, we now refer to this subset as CLEC4-KCs and have removed terms related to "identity loss" in the revised manuscript.

To address comments 1 and 2, we have conducted a new scRNA-seq analysis on FACS-sorted F4/80^{hi}CD11b^{int}CD64⁺CLEC4F⁻TIM-4⁺ cells (C-T⁺) from 42 days infected mouse liver samples. Starting with approximately 40 million leukocytes, we obtained around 16,000 CLEC4F⁻TIM-4⁺ cells. After sorting, about

6,000 live cells were counted and used for downstream scRNA-seq, with 429 cells detected by Illumina sequencing following the 10X Genomics pipeline; of these, 107 passed our QC filters. This extreme cell loss reflects the same challenge encountered in our initial scRNA-seq, where a specific cluster for these cells was undetectable.

A pseudobulk comparison of C-T⁺ sorted cells with the clusters identified in our previously sequenced samples, showed that over half shared a similar gene expression profile with resKCs and proliferating cells, and approximately 20% were grouped with transitioning monocytes (Supplementary Fig.6d). The C-T⁺ cells shared the expression of several KC-associated transcription factors (Supplementary Fig.6e). For KC identity genes, as defined by Williams et al. 2022, the transcriptional profiles of resKCs and C-T⁺ cells were highly similar (Supplementary Fig.6f). Functionally, C-T⁺ cells showed lower expression of *Ccl24* than resKCs but the highest levels of *Il10* and *Il1a* (Supplementary Fig.7a). Collectively, these findings, along with our previous confocal imaging data, parabiosis, and *Ccr2*^{-/-} and *Clec4f*^{Cre-TdT}*ZsGreen* mice suggest that C-T⁺ cells are KCs within granulomas that maintain the expression of many resKCs genes, downregulate CLEC4F on their surface, and show increased expression of some pro-inflammatory chemokines, iNOS, and IL-10.

2. Indeed, in this regard, in their scRNA-seq analysis the authors could not convincingly identify these exKCs, suggesting they either did not isolate them well or that they were perhaps in the transitioning monocyte fraction. Given that the authors find CLEC4F-TIM4⁺ cells in their flow analysis (extended data Fig 1) it seems unlikely that they are not isolating these cells for the scRNA-seq analysis. Additionally, the idea that a KC would lose its identity and regain monocyte markers seems a bit far-fetched. Overall, the authors need to better assess where these cells lie in their scRNA-seq analysis. This could be done in multiple ways. Given that the cells are there in their flow cytometry data, the authors could do CITE-seq analysis using CLEC4F and TIM4 barcoded antibodies to then find back the cells and assess their transcriptional profile. Perhaps even easier, the authors could just sort the CLEC4F-TIM4⁺ cells from the liver and run them by either scRNA-seq or even bulk RNA-seq and then overlay the signatures on their current data.

See response to comment #1.

3. The annotation of the scRNA-seq analysis is based on published data from a NAFLD model, exactly how the annotations were transferred here is not clear, for example if MAC1 do not express *Clec1b* how similar are they really to the published MAC1s? What other elements of the signature do they share? Here a table with DEGs per cluster would have been very useful for comparison. The same also holds for the Trans mono 1 cluster.

We have revised the method description of the annotation transfer analysis in the manuscript:

Lines 649-655:

“To be consistent with KC annotation from the literature, we mapped our data onto a reference scRNA-seq dataset from Remmerie et al. Initially, anchor point cells were selected between the reference and our dataset query using the *FindTransferAnchors* function, which is based on shared gene expression and mutual nearest neighbors (MNNs) in a shared low-dimensional space. Next, we used those anchors to assign cell type labels and project the reference dimensional reduction onto our dataset using the *MapQuery* wrapper.”

The list of differentially expressed genes in each transcriptomic cluster identified in both our scRNA-seq have been added to Supplementary Tables 1 and 2.

4. The authors claim that the KCs migrate out of the sinusoids to form granulomas, but this claim is not substantiated. Could it not equally be that these granulomas form in place and damage the sinusoids, leading to their loss? Indeed, the areas where they are present show no sinusoidal vessels but large areas like this do not exist in healthy liver. I am convinced that the granulomas are not in sinusoids at this timepoint, but the idea that the KCs are really migrating that is not conclusively shown. To really claim migration, the authors would need to demonstrate KC migration with intravital imaging or a confocal timecourse. Perhaps looking at an earlier timepoint (or a few) after infection would help here.

Following the reviewer’s suggestion, we have performed live imaging of *Clec4f^{Cre-TdT}ZsGreen* mice at an earlier time point after infection, specifically at 19 d.p.i. Consistent with our immunofluorescence findings shown in Fig.1, early granulomas were tdTomato⁺ZsGreen⁺ (Fig.3b,d), indicating that CLEC4F expression is still detectable within early-stage granulomas, and further supporting the KC origin of granuloma cores.

KC redistribution during VL has been demonstrated by the administration of nanobeads before infection as a method to track the KCs, which were observed to subsequently cluster at the core of the granulomas (Beattie et al., PLoS Pathog 2010. PMID: 20300603), suggesting that KCs migrate to form granulomas. We did not observe active migrating KCs in our live imaging at 19- or 42-d.p.i. A failure to observe KCs migration during mycobacterial infection was reported by Egen et al. (Immunity, 2008. PMID: 18261937), where they suggested that KC redistribution to form granulomas occurred gradually over a period longer than could be captured by the 1-hour video recordings.

We have performed a series of experiments, two included in the original manuscript and one added to the revision, to address the important issue of

sinusoidal damage that may account for the apparent redistribution of KCs in the perisinusoidal space. By using CD31 to label the sinusoids in *Clec4f^{Cre-TdT}ZsGreen* mice, we observed that *L. infantum* infection caused changes in the sinusoidal network, which were more pronounced at 42 d.p.i. compared to 19 d.p.i. (Fig.3e). At 19 d.p.i., small early clusters were identified at least partially outside the sinusoids and did not alter sinusoid distribution or integrity (Fig.3f and Supplementary movie 1). Changes in the sinusoidal network became evident around larger ZsGreen clusters, which were mostly outside the blood vessels (Fig. 3g and Supplementary movie 2). The sinusoids appeared to be displaced, pushed outward and downward by the expanding granuloma in the perisinusoidal space (Fig.3g, arrows and Supplementary movie 3). Importantly, there were no signs of damage to the vessel wall or changes in blood vessel diameter (Fig. 3g and Supplementary movie 3).

In a new experiment to further investigate if the formation of KC clusters was associated with damage or rupture to the sinusoids, we stained red blood cells (RBCs) with Ter119 and performed live imaging in WT mice at 3- and 6-w.p.i. RBCs were observed flowing within the sinusoids, but without evidence of leakage into granulomas spaces at either time point (Supplementary movies 4 to 7). Consistently, tracking of RBCs revealed their movement within the sinusoids but not in regions containing granulomas (Supplementary Fig.3a). We did, however, detect RBCs within some F4/80⁺ clusters at 3 w.p.i. (Supplementary movies 4 and 5) and in areas corresponding to granulomas at 6 w.p.i. (Supplementary movies 6 and 7). However, unlike the dynamic streaming of RBCs observed within the sinusoids, these RBCs were mostly stationary. Additionally, static RBCs were observed in association with several F4/80⁺ cells inside the sinusoids (Supplementary movie 7), which may account for the presence of the stationary RBCs in granulomas, potentially carried there by KCs redistribution during granuloma formation. Additionally, a previous study has demonstrated that KC depletion using clodronate can temporarily disrupt the sinusoidal layer, leading to structural gaps (Baet et al., Cell Microbiol, 2007. PMID: 16953803). Similarly, the redistribution of KCs outside the sinusoids may contribute to the formation of such gaps, further accounting for the accumulation of RBCs in granulomas. Lastly, most larger clusters of ZsGreen⁺ cells at 19- and 42-d.p.i. did not stain following i.v. anti-F4/80 (Fig.3h), suggesting a loss of contact with the sinusoids, and little or no vascular leakage into the perisinusoidal space.

Taken together, the new data from 19-day infected *Clec4f^{Cre-TdT}ZsGreen* mice and live imaging of RBCs showing no evidence of damage or rupture to the blood vessels, does not support the idea that granulomas form in place and damage the sinusoids. These findings, along with our data from infected *Clec4f^{Cre-TdT}Confetti* mice showing that KC granulomas can be polyclonal, suggest that granuloma expansion occurs following KC redistribution outside the sinusoids. This process happens without signs of blood vessels rupture, but physically displaces the sinusoidal network.

5. While the data is convincing that the CLEC4F-TIM4⁺ exKCs and CLEC4F⁺TIM4⁺

KCs are resident and the other macrophage populations monocyte derived based on the data provided, I find the CCR2KO data somewhat confusing and requiring further explanation. It is well known that even in the absence of CCR2 some monocytes/monocyte precursors can egress the BM and thus it has previously been shown that in a KC depletion model CCR2KO mice still develop moKCs but with a slightly slower kinetics as the few monocytes need to proliferate more to fill the niche (Bonnardel et al., Immunity 2019). Thus, here I question why there are suddenly no moKCs at all? One possibility is that the remaining ResKCs kick into proliferation in absence of monocytes to refill the niche as was shown in a partial KC depletion model (Scott et al., Nat Comms 2016). But this would suggest that niche repopulation in steady state also occurs via the two mechanisms proliferation and monocyte recruitment. Thus, the authors should check this in their model. Do resKCs proliferate (scRNA-seq data would suggest yes), and is this proliferation enhanced in CCR2KO mice? Do the exKCs also proliferate in the granulomas? With this in mind, could it be the proliferation rather than the loss of monocytes contributing to increased parasite load in CCR2KO mice? Here I think the claims should be toned down as with so much changing in the CCR2KO mice, I don't think the authors can be sure that this is due to lack of monocytes?

Following the reviewer's suggestion, and to minimize cell isolation bias during digestion and Percoll purification, we used stored tissue samples from our infected WT and *Ccr2*^{-/-} mice and stained them with anti-Ki67 to identify proliferating cells. No differences in cell proliferation were observed between WT and *Ccr2*^{-/-} naïve mice (panel a, below).

In 42 day-infected WT mice, among the proliferating F4/80⁺ cells, the majority were the CLEC4F⁻TIM-4⁻ momacs, while resKCs corresponded to 8%, and CLEC4F⁻TIM-4⁺ cells to 30% of the proliferating cells (Supplementary Fig.2c,d). Thus, proliferation might also support increased frequencies of CLEC4F-KCs cells within WT granulomas.

In infected *Ccr2*^{-/-} mice, proliferation was increased 4.6-fold in resKCs, and 1.7-fold in CLEC4F-TIM-4⁺ cells compared to WT mice (Supplementary Figs.2c,d). The results align with the reviewer's suggestion that reduced monocyte infiltration in infected *Ccr2*^{-/-} livers is compensated by increased proliferation of the remaining KCs, particularly the CLEC4F⁺TIM-4⁺ resKCs. This suggests that increased proliferation of KCs might contribute to the higher parasite load in

Ccr2^{-/-} mice. It should be noted, however, that infected *Bach1*^{-/-} mice (see response to comment 6 below) also showed increased resKCs proliferation, but did not exhibit differences in parasite loads, indicating that increased resKC proliferation alone is not sufficient to promote parasite growth. The following has been added to the revised manuscript (lines 341-346): “The contribution of momacs to hepatic resistance is likely due to their cooperative functions with CLEC4F-KCs in the effector response. However, the increased proliferation of resKCs observed in infected *Ccr2*^{-/-} mice (Supplementary Fig.2c) suggests that monocyte recruitment into the open niche might also serve to limit the homeostatic proliferation of resKCs, which better support parasite survival and growth. In either case, a heterogeneous population containing both KCs and momacs is important for effective control of VL”.

Lines 358-360: “In infected *Ccr2*^{-/-} mice, impaired monocyte recruitment, increased resKCs proliferation and reduced granuloma macrophage heterogeneity were associated with higher liver parasite burdens”.

Discussion of these findings has been included on lines 456-468.

6. The authors also claim that resKCs die by ferroptosis leading to their replacement by moKCs. This is shown using *Bach1*KO mice as *Bach1* deficiency is linked with decreased ferroptosis. However, again as this is a full KO mouse, I am not sure this is the only explanation for the data. Could it not also be that *Bach1* deficiency interferes with monocyte recruitment/differentiation? How do monocytes look in the livers of these mice, are they still recruited? Do ResKCs proliferate more in this model? Could that explain the reduced need for moKC differentiation (see pt 5 above). After checking these aspects, it would also be interesting for the authors to look earlier post infection. Now they check D42 pi when moKCs are present thus the niche is already refilled (at least partially) perhaps checking earlier would allow them to better visualize the dying KCs? Although I certainly acknowledge that capturing cell death in vivo is not straight forward.

We agree that additional factors/mechanisms may contribute to the phenotype observed in *Bach1*^{-/-} mice. Regarding BACH1's potential role in monocyte recruitment to the liver, we have added the relevant data to Supplementary Fig.4i, showing no differences in monocyte infiltration comparing livers of infected WT and *Bach1*^{-/-} mice. We also examined cell proliferation and found a 2.9-fold increase in resKCs proliferation in infected *Bach1*^{-/-} compared to WT mice (Supplementary Fig.5j). No proliferation differences were observed among other F4/80⁺ or F4/80⁻ cells (Supplementary Fig.5j,k). Notably, the increased resKCs numbers did not affect parasite loads (Supplementary Fig.5l). Regarding the suggestion to address cell death at an earlier time point, the data for lipid peroxidation and GSH levels at 19 d.p.i. (and 73 d.p.i.) are shown below. At 19 d.p.i. both TIM-4⁺ (below, panel a) and TIM-4⁻ (panel b) populations in *Bach1*^{-/-} mice showed lower lipid peroxidation compared to infected WT mice. At 73 d.p.i., *Bach1*^{-/-} mice also had reduced GSH levels relative to infected WT mice (panel c). Parasite loads were similar between strains at both time points (panel d). These

results, along with 42 d.p.i. data (Supplementary Fig.5), suggest that BACH1 regulates resKCs numbers through proliferation, lipid peroxidation, and potentially ferroptosis during the peak of the immune response at 42 d.p.i., without impacting parasite loads.

Minor Concerns:

1. The gating strategy in extended figure one, likely misses many macrophages (monocyte derived) that are not Cd11bint. Can the authors also show these gates in the infected mice and specify for the scRNA-seq analysis exactly what was sorted?

We have added the gating information for infected mice and CD11b⁺ macrophages, referred to as monocyte-derived cells (mo-cells), to Supplementary Fig.1a. The gates used for sorting cells in the first scRNA-seq are highlighted in green, while those used for the second scRNA-seq experiment are marked with both solid and dashed green (CLEC4F⁻TIM-4⁺).

2. Why are there no capsule macrophages annotated in the scRNA-seq data?

Remmerie et al. (Immunity, 2020. PMID: 32888418) employed a different macrophage isolation protocol, using collagenase A and centrifugation to separate hepatocytes from other cells. By contrast, we used collagenase IV and Percoll gradient to isolate leukocytes only. Since Remmerie et al. identified only a small number of capsule macrophages, these cells may have been lost during our digestion and/or Percoll purification and were not identified in our scRNA-seq analysis.

Reviewer #2 (Remarks to the Author):

In their manuscript „Kupffer cell and recruited macrophage heterogeneity orchestrate granuloma maturation and hepatic immunity in visceral leishmaniasis“, Pessenda et al. address the question of how the myeloid network in the liver is interacting and replenished in the context of *L. infantum* infection. Using a series of very elegant reporter system approaches which address the dynamics of Kupffer cells and monocyte-derived cells, the authors show evidence that during visceral Leishmaniasis, the resident Kupffer cell (resKC) niche is emptied through detachment from the liver sinusoids and relocalization to granulomas, as well as through ferroptotic cell death. In particular, the authors claim that resKC lose CLEC4F expression upon detachment from Liver sinusoidal endothelial cells (LSECs) and aggregation within granulomas, and, together with transitioning monocytes, exhibit an inflammatory expression profile, while resKC as well as monocyte-derived KC outside the granuloma show a homeostatic or immunoregulatory path. By employing *Ccr2*-deficient animals, the authors finally provide results that underline the importance of recruited monocyte for pathogen control.

Overall, this is a very interesting study which deserves attention from both infection immunologists as well as from the myeloid cell field. The experiments are beautifully designed for the scientific questions investigated, and the results fully support the conclusions drawn by the authors. While the characterization of the different KC and monocyte-derived cell turnover, population dynamics and localization, is admirable, the manuscript could be strengthened if some of the mechanistic aspects that the authors touch upon in the second part of the manuscript would be addressed more thoroughly (see below).

Major comments:

1. In Figure 3, the authors show evidence for the detachment of T-C⁺ KC from LSECs and hypothesize in accordance with Bonnardel et al, (*Immunity* 2019) that the lack of contact between KCs and LSECs might be responsible for their loss of identity, i.e. CLEC4F downregulation. In the said reference, DLL-Notch signaling is shown to be a main determinant of KC identity. Is the stimulus from the LSECs indeed continuously required, i.e. would blocking antibodies against DLL1 and DLL4 in the *L. infantum* model also result in enhanced identity loss of the KCs?

We appreciate this suggestion, and we conducted the proposed experiment with results shown below. Since we observed no significant changes in macrophage composition during early infection, we treated WT mice twice weekly with 0.2mg of anti-DLL1 and 0.2mg of anti-DLL4 during weeks 4 and 5 post-infection. This treatment reduced the frequency and number of CLEC4F-TIM-4⁺ cells within granulomas (below, panels a-b,e), and also decreased granuloma volume (panels c,f). However, these changes did not impact parasite loads at 42 d.p.i. (panel d). The results do not support that blocking DLL-Notch signaling drives CLEC4F downregulation within granulomas.

Because we observed differences in granuloma composition after 2 weeks of treatment, we tried the same treatment schedule over 4 weeks, from weeks 2 to 5 post-infection, to assess if prolonged changes in granuloma composition would affect parasite loads. Extended treatment, however, did not alter KC composition within granulomas (panels g-h), granuloma volume (i), or parasite loads (j). Given the inconsistent results, and the number of experiments needed to optimize the treatment regimen, we have chosen not to include these findings in the revised manuscript.

2. In their scRNAseq experiment, the authors identify inflammatory, granuloma-associated macrophages via the expression of iNOS, which they have shown using confocal imaging to be limited to granulomas. Also, they note that they could not identify a defined cluster of CLEC4F-Tim-4+. Could Leishmania infection, which the authors show nicely in their confocal microscopy, be used to characterize the infection rates in C+T+, C+T-, C-T+ and C-T- macrophages by flow cytometry to underline the observations from histology? Related to this, could the *L. infantum* RNA sequencing reads (which should be available from the 10x data) be mapped to the UMAP plots to

characterize and assign infected cells to the different populations according to the *L. infantum* infection rate?

The challenges in recovering KCs (see response to comment 1 from rev#1), particularly the CLEC4F-TIM-4⁺ population, which we believe results in the overrepresentation of monocyte-derived cells in our flow cytometry preparations, led us to use confocal imaging of tissue sections as it provides a more accurate assessment of parasite distribution among the specific macrophage subsets. Regarding identifying *L. infantum* reads in our scRNA-seq dataset, only 35 cells showed *L. infantum* reads mapped to either nuclear or mitochondrial genomes above background noise (17 in the first sorted samples and 18 in the newly sorted CLEC4F-TIM-4⁺ samples). After filtering, only 12 cells remained, making it uninformative to assign these reads to different populations. This result might also indicate that the infected cells are more susceptible to cell death and loss during sample isolation, sorting, and processing.

3. The authors use *Ccr2*-deficient mice which lack monocyte recruitment to demonstrate the importance of their findings for control of *L. infantum* and show that while the resKC still express iNOS under these conditions, this seems not to be sufficient to contain the infection. Moreover, IL-4 in the liver seems to be increased, which the authors conclude could additionally impair pathogen control in *Ccr2*-deficient mice. Could anti-IL-4 treatment at least partially rescue the phenotype? Alternatively, could a mixed *Ccr2*-DTR : iNOS^{-/-} bone marrow chimera approach be used to enable only the recruitment of iNOS^{-/-} monocytes, thus dissecting the influence of resKC-produced from monocyte-produced nitric oxide?

We thank the reviewer for this suggestion. In visceral leishmaniasis, IL-4 has been shown to play a role in controlling *L. donovani* in the livers of infected BALB/c mice at 30 d.p.i. (Stager et al., *Infect Immun*, 2003. PMID: 12874364), but it does not appear to affect parasite control in the livers of C57BL/6 mice (Satoskar et al., *Infect Immun*, 1995. PMID: 7591152). Similarly, we observed no differences in parasite loads in the livers of *L. infantum* infected C57BL/6 IL-4^{-/-}/13^{-/-} mice at various times post-infection (below, panel a), or in IL-4^{-/-} at 19 d.p.i. (panel b). To avoid the possible pleiotropic effects of the global IL-4 deficiency, and to specifically assess the role of IL-4 during granuloma development, as suggested by the reviewer, we treated *Ccr2*^{-/-} mice with 0.4mg of anti-IL-4, administered 3 times per week during weeks 4 and 5 post-infection. This treatment did not impact parasite loads (panel c), frequency of KC subsets (panels d-e), volume (panel f) and number of granulomas (panel g-h). Since we did not observe any differences following anti-IL-4 treatment, we decided not to include these results in the revised manuscript.

However, we have added the complete Luminex data for 42-day infected WT and *Ccr2*^{-/-} mice to Supplementary Fig.7g. The data revealed a markedly different cytokine/chemokine environment between infected WT and *Ccr2*^{-/-} mice, which may explain why IL-4 neutralization alone was insufficient to produce an effect.

Consequently, we have removed our previous comment suggesting that increased IL-4 could be impairing pathogen control in *Ccr2*^{-/-} mice.

A description of the Luminex data has been added to lines 337-340, and a discussion of susceptibility of the *Ccr2*^{-/-} mice has been included on lines 341-346 and 456-468 in the revised manuscript.

Minor points:

Figures 2J, 2K and 3c: The tdTomato fluorescence is hardly recognizable, possibly due to the resolution. This might be improved by showing the single channel images not for the whole images, but (also) for the magnification insets.

We have increased the contrast of the tdTomato images and updated Figure 3.

Line 117 use past tense: “the few that we detect”

Line 229 sentence hard to understand: “the Nos2 annotated UMAPs will have identified macrophages”

Text modifications have been made.